

# Contribution of blowing snow sublimation to the surface mass balance of Antarctica

Srinidhi Gadde [1,2] and Willem Jan van de Berg [1]

[1]Institute for Marine and Atmospheric Research, Utrecht University, Utrecht, The Netherlands
[2]Faculty of Geo-Information and Earth Observation (ITC), Univeristy of Twente, Enschede, The Netherlands

**Correspondence:** s.nagaradagadde@utwente.nl, w.j.vandeberg@uu.nl

**Abstract.** Blowing snow sublimation is an important boundary layer process in polar regions and is the major ablation term in the surface mass balance (SMB) of the Antarctic ice sheet. In this study, we update the blowing snow model in the Regional Atmospheric Climate Model (RACMO), version 2.3p3, to include, among other things, the effect of blowing snow sublimation in the prognostic equations for temperature and water vapour. These updates are necessary to remove undesired numerical
artefacts in this version's modelled blowing snow transport fluxes. Specifically, instead of a uniformly discretised ice particle radius distribution used in the previous version of the model which limited the maximum ice particle radius to $\leq 50~\mu$m, we use a non-uniformly discretised ice particle radii to include all relevant ranges of radii between 2 to 300 $\mu$m without any additional computational overhead. The updated model results are compared against the meteorological observations from site D47 in Adélie Land, East Antarctica. The updates alleviate the numerical artefacts observed in the previous model results and
successfully predict the power-law variation of the blowing snow fluxes with wind speed while improving the prediction of the magnitude of the blowing snow fluxes. Furthermore, we obtain an average blowing snow layer depth of $230 \pm 116$ m at the observation site D47, which matches well with the typical values obtained from the satellite observations. A qualitative comparison of the blowing snow frequency from updated RACMO with CALIPSO satellite observations shows that RACMO successfully predicts the blowing snow frequency. For the period 2000 – 2010, compared to the previous model version, the
contribution of integrated blowing snow sublimation is increased by 30%, with a yearly mean of $176 \pm 4~\mathrm{Gt\,yr^{-1}}$. It contributes to a 1.2% reduction in the integrated SMB of the Antarctic ice sheet compared to the previous model results. In addition, we observe significant changes in the sublimation in coastal and lower escarpment zone, indicating the importance of the model updates to the climatology of blowing snow in Antarctica.

## 1 Introduction

In the coastal regions of Antarctica, strong katabatic winds lift loose snow off the ground, causing drifting snow (e.g. Kodama et al. (1985)). When this snow rises further and is suspended in the atmospheric boundary layer, it is designated as blowing snow. This wind-driven transport can be categorised as drifting ($< 1.8$ m a.g.l) and blowing ($> 1.8$ m a.g.l) snow (Serreze and Barry, 2005, p. 54). It redistributes the snow on the surface of an ice sheet and can also give rise to blue-ice areas, affecting the local surface energy balance (SEB) (van den Broeke and Bintanja, 1995). Furthermore, it is well known that the suspended snow





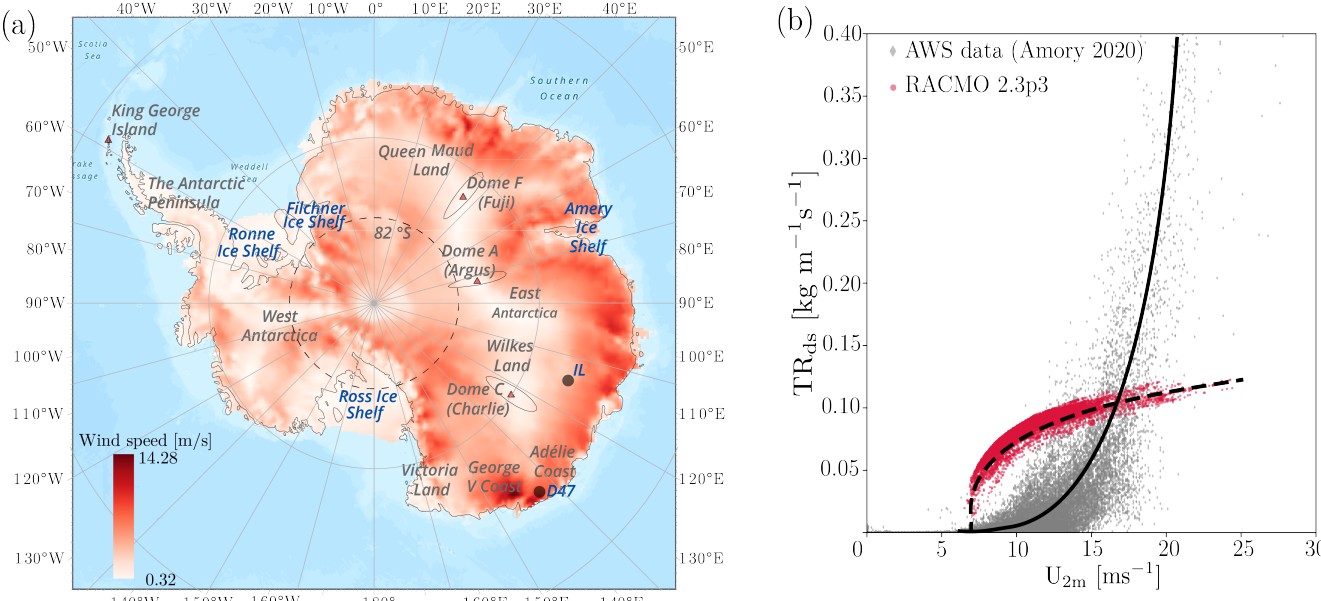

**Figure 1.** (a) Yearly average (2000-2010) 10–m wind speed $(\mathrm{m\,s^{-1}})$. Location of observational site D47 in Adélie Land, East Antarctica, IL represents an interior location (71.1°S, 111.7°E), and dashed lines represent the latitude 82°S, North of which CALIPSO satellite data is available. (b) Variation of near surface blowing snow flux $\mathrm{TR_{ds}}$ $(\mathrm{kg\,m^{-1}\,s^{-1}})$ vs 2–m wind speed $(\mathrm{m\,s^{-1}})$. Solid and dashed lines represent the variation of observed and simulated (RACMO2.3p3) near surface snow-drift fluxes, respectively. Note that RACMO2.3p3 fails to predict the variation of blowing snow transport reasonably.

particles are more prone to sublimation than surface snow (Schmidt, 1972; Bintanja, 2001). Therefore, drifting and blowing snow transport and sublimation are important factors contributing to Antarctica's surface mass balance (SMB), particularly in coastal regions (Bintanja, 1998). For brevity, from hereon, both drifting and blowing snow are combined and referred to as blowing snow.

     Blowing snow is a significant contributor to the (local) SMB of the polar regions and plays a crucial role in the climate
system of Antarctica. While there have been automatic weather station (AWS) observations of blowing snow-related processes from Antarctica (van den Broeke et al., 2004; Thiery et al., 2012; Barral et al., 2014; Amory, 2020), continent-wide estimates of blowing snow are difficult to obtain from observations. Continent-wide estimates derived from satellite-based products are available (Palm et al., 2017). However, they are restricted to optically thin cloud conditions and snow suspended in upper layers of the boundary layer (> 30 m a.g.l.) (Palm et al., 2011) and therefore are not suitable for estimates of near-surface blowing
snow and its contribution to SMB. Hence, these continent-wise estimates can only be obtained by parameterising blowing snow processes and embedding these parameterisations in regional climate models (RCMs) (Bintanja, 1998; Déry and Yau, 2001; Lenaerts and van den Broeke, 2012; Amory et al., 2021; Toumelin et al., 2021). However, the representation of blowing snow in RCMs is challenging due to the complex and dynamic nature of this phenomenon involving multiple feedbacks with related processes such as snow precipitation and surface sublimation.





Including a blowing snow model in regional climate models (RCMs) has been found to improve the SMB estimates in the regions where katabatic winds form (Mottram et al., 2021). Specifically, without modelling blowing snow processes, it is difficult to capture the spatial gradients in the sublimation of snow accurately (Agosta et al., 2019), which is particularly important in the escarpment regions of Antarctica. To improve our understanding of the Antarctic climate, it is crucial to accurately model the occurrence and impacts of blowing snow in RCMs. However, due to the coupled nature of blowing snow

and the high sensitivity of the model to parameters, it is difficult to obtain a perfect agreement between observed and RCM estimates of blowing snow fluxes (Lenaerts et al., 2014; van Wessem et al., 2018; Amory et al., 2015, 2021).

The polar version of the regional atmospheric climate model (RACMO) (van Wessem et al., 2018; van Dalum et al., 2022) is coupled with a blowing snow scheme based on the PIEKTUK model (Déry and Yau, 2001; Lenaerts et al., 2012) to represent snow transport in polar regions. Previously, in the absence of near-surface measurements of blowing snow fluxes, only blowing

snow frequencies from RACMO were evaluated against the observations (Lenaerts et al., 2012). Further evaluations of RACMO against snow particle counter (SPC) observations from Greenland showed that RACMO2.3p1 overestimated the snow particle transport when direct comparisons of fluxes were made (Lenaerts et al., 2014). In RACMO2.3p2 (van Wessem et al., 2018), the linear saltation coefficient was subsequently halved to match these blowing snow fluxes.

Recently, we evaluated blowing snow fluxes from RACMO against the SPC observational data by Amory (2020) at site

D47 (location: 67.4°S, 138.7°E), Adélie Land, East Antarctica. Figure 1(a) shows the yearly (2000-2010) average 10-m wind speed obtained by RACMO and the location of the observation site D47. Since the coastal regions of Antarctica witness very-high speed winds (Fig. 1(a)) and the concentration of blowing snow particles increases with the wind speed (Radok, 1977; Budd, 1966; Amory, 2020), the blowing snow transport ($TR_{ds}$ $kg\,m^{-1}s^{-1}$) is expected to increase in a power-law fashion with velocity. However, Figure 1(b) shows that $TR_{ds}$ from RACMO does not show a rapid increase with velocity as

expected. Evaluation of RACMO blowing snow fluxes with observations also showed that RACMO consistently underperforms in accurately predicting the magnitude of the observed fluxes. The evaluation shows the need to improve the blowing snow model in RACMO and systematic comparison of blowing snow fluxes against observations to obtain reliable estimates of Antarctic SMB.

In this study, several updates to the blowing snow scheme in RACMO are presented. The updates aim to improve the

coupling of the blowing snow processes with RACMO atmospheric physics. Next, near-surface blowing snow fluxes obtained from RACMO are compared against the observed fluxes from site D47, Adélie Land, East Antarctica (Amory, 2020). The observations from site D47 are particularly suitable for evaluations since the region experiences frequent blowing snow, and the observations employ $2^{nd}$ generation FlowCapt$^{TM}$ sensors, which have been found to predict the blowing snow fluxes with reasonable accuracy (Amory, 2020). The details of RACMO and the modifications to the blowing snow scheme in RACMO

are presented in Section 2, and details of the observational site and available data are presented in Section 3. Blowing snow frequency and fluxes from RACMO are evaluated against the observations in Section 4, followed by comparing results against the original version of RACMO. Furthermore, we discuss the impact of the snow drift updates on the continent-wide estimates of SMB for Antarctica by comparing the modelled SMB for 2000-2010, followed by conclusions in Section 5.





## 2    Model descriptions

### 2.1    Regional Atmospheric Climate Model (RACMO)

RACMO is built on the semi-implicit semi-Lagrangian dynamics kernel of the numerical weather prediction model HIRLAM (High-resolution limited area model; Undén et al. (2002)), version 5.0.3, with the European Center for Medium-Range Weather Forecasts (ECMWF) physics package, including both surface and atmospheric processes, from cycle 33r1 (ECMWF, 2009). The model assumes hydrostatic equilibrium, and the operational polar version (version 2.3p2) has been verified to produce realistic results at the resolutions used in this study (van Wessem et al., 2015, 2016). This polar (p) version of RACMO2 includes a multilayer snow model that calculates the snow albedo evolution, melt, refreezing, percolation and run-off of meltwater (Greuell and Konzelmann, 1994; Ettema et al., 2010; Kuipers Munneke et al., 2011). It also includes a blowing snow scheme based on the PIEKTUK model (Déry and Yau, 1999; Lenaerts et al., 2012).

In the newer version of RACMO2, version 2.3p3, hereafter abbreviated as Rp3, the snow and ice albedo parameterisations were updated using Two-streAm Radiative TrasnfEr in Snow model (TARTES; Libois et al. (2013)) coupled with the Spectral-to-NarrOWBand ALbedo (SNOWBAL) module, version 1.2 (van Dalum et al., 2019). Rp3 has produced results that compare well with both in-situ and remote sensing observations of SMB of Antarctica (van Dalum et al., 2022). RACMO2 and Rp3 are introduced in detail in Noël et al. (2018) and van Dalum et al. (2019), respectively. At the lateral boundaries, the simulations presented here are forced with ECMWF ERA5 reanalysis data (Hersbach et al., 2020) with an update interval of 3 hours.

### 2.2    Blowing snow model

In RACMO, we use the bulk (non-spectral) version of the PIEKTUK model (Déry and Yau, 1999), which employs an evolution equation for the mixing ratio of blowing snow $q_b$ $(\mathrm{kg\,kg^{-1}})$ and an additional equation for the evolution of snow particle number concentration $N$, which is the double-moment version of the PIEKTUK model (PIEKTUK-D, Déry and Yau (2001)). PIEKTUK is an Inuktitut word for blowing snow (Déry et al., 1998). Here, we introduce only the essential features of the PIEKTUK model, and the additional details can be found in Déry and Yau (2001).

Figure 2 shows the blowing snow processes and the coupling between PIEKTUK and RACMO, presenting the important snow transport mechanisms over an ice sheet. When the friction velocity, a measure of the wind shear at the surface, exceeds the threshold friction velocity, the snow particles perform a downwind motion of a series of jumps or skips, a process called saltation. When the saltating snow particles get suspended in the boundary layer due to turbulent mixing, they form the blowing snow. In PIEKTUK, this transition from saltation to suspension, governed by different physical mechanisms, is assumed to happen at the elevation $h_{\mathrm{salt}}$. It is worth mentioning here that, in RACMO, the suspension is coupled with saltation, as saltation does not contribute to long-distance snow transport; the model is only activated when the wind is energetic enough to facilitate suspension. For the blowing snow governing equations, the saltation flux parameterisation serves the boundary condition. In Section 2.2.1 we introduce the saltation flux parameterisation followed by the evolution equations of blowing snow in Section 2.2.2.



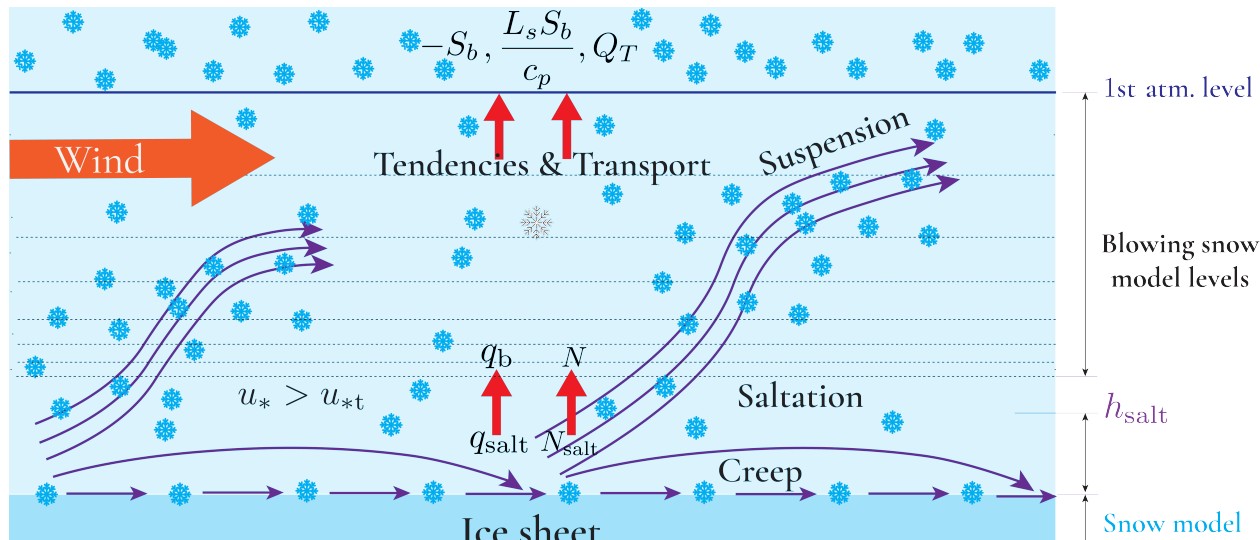

**Figure 2.** Schematic showing the blowing snow model levels, RACMO model levels, and key processes involving blowing snow. $Q_T$ represents the snow transport due to blowing snow. The figure shows that the model level of blowing snow is above the saltation height. In the schematic, $q_b$ (mixing ratio) and $N$ (number concentration) represent the boundary conditions for the blowing snow model calculated from $q_{salt}$ and $N_{salt}$ using the classical equation for suspended particle concentration.

### 2.2.1 Saltation flux

There exist several empirical formulations for erosion of the snow particles in the saltation layer, $q_{salt}$ $(\mathrm{kg\,kg^{-1}})$. In previous RACMO versions, it has been parameterised using Déry and Yau (1999),

$$q_{salt} = c_{salt}\left(1 - (u_{thr}/U_{fml})^{2.59}\right), \tag{1}$$

in which $c_{salt}$ is a constant, initially set to $0.385$ and retuned to $0.192$ in RACMO versions 2.3p2 and Rp3, respectively. Furthermore, $u_{thr}$ a threshold wind velocity $(\mathrm{m\,s^{-1}})$, and $U_{fml}$ the wind speed at the first model level $(\mathrm{m\,s^{-1}})$. The threshold wind velocity is defined by $u_{thr} = 9.43 + 0.18\,T_{2m} + 0.0033\,T_{2m}^2$ with $T_{2m}$ in °C (Déry and Yau, 1999).

In this study, we update the saltation parameterisation with an alternative empirircal parameterisation, proposed by Pomeroy (1989), is

$$q_{salt} = \frac{e_{salt}}{gh_{salt}}\left(u_*^2 - u_{*t}^2\right), \tag{2}$$

where $e_{salt}$, the saltation efficiency, is set to $1/(3.25u_*)$, $u_*$ $(\mathrm{m\,s^{-1}})$ is the friction velocity, $h_{salt} = 0.08436u_*^{1.27}$ represents the thickness of the saltation layer (m) according to the relation of Pomeroy and Male (1992), $g = 9.81$ is the gravitational acceleration $(\mathrm{m\,s^{-2}})$, and $u_{*t}$ represents the threshold friction velocity $(\mathrm{m\,s^{-1}})$.

The parameterisation of the threshold friction velocity in Equation (2) is given by Gallée et al. (2001):

$$u_{*t} = u_{*t0}\exp\left(\frac{-n}{1-n} + \frac{n_0}{1-n_0}\right), \tag{3}$$





where, $n = (1 - \rho_{\mathrm{s}}/\rho_{\mathrm{i}})$ is the snow porosity, $n_0 = (1 - \rho_0/\rho_{\mathrm{i}})$, with $\rho_{\mathrm{s}}$ as the actual mean snow density of the upper 5 cm, and $\rho_{\mathrm{i}}$ is the density of ice. $n_0$ is the porosity of fresh snow with $\rho_0 = 300\ \mathrm{kg\,m^{-3}}$. The reference threshold friction velocity $u_{*\mathrm{t}0}$ is calculated based on the potential for snow erosion by the wind, which is characterised by a snow mobility index, Mo, which is given by the relation, $\mathrm{Mo} = 0.75d - 0.5s + 0.5$, where the variables $d$ and $s$ represent the snow grain dendricity, and sphericity,

respectively. However, dendricity and sphericity are not modelled in RACMO. Therefore, we take $d = s = 0.5$, hence set Mo to 0.625. Finally, $u_{*\mathrm{t}0}$ is defined by Gallée et al. (2001) as

$$u_{*\mathrm{t}0} = \frac{\log(2.868) - \log(1 + \mathrm{Mo})}{0.085} \, C_D^{0.5}, \tag{4}$$

where $C_D = u_*^2/U_{\mathrm{fml}}^2$ represents the drag coefficient of momentum.

### 2.2.2 Evolution of blowing snow

To calculate the sublimation and transport of blowing snow, the evolution equation for the blowing snow mixing ratio $q_b$ $(\mathrm{kg\,kg^{-1}})$ is written as:

$$\frac{\partial q_b}{\partial t} = \frac{\partial}{\partial z}\left(K_b \frac{\partial q_b}{\partial z} + v_b q_b\right) + S_b \tag{5}$$

where t (s) denotes time, z (m) the vertical coordinate, $v_b\ (\mathrm{m\,s^{-1}})$ is the bulk terminal velocity, $K_b\ (\mathrm{m^2\,s^{-1}})$ represents turbulent eddy diffusivity for blowing snow, and $S_b\ (\mathrm{kg\,kg^{-1}\,s^{-1}})$ the bulk sublimation rate. This equation is implicitly discretised in

the vertical direction and solved using the tri-diagonal matrix algorithm. The lowest blowing snow model level is set to 0.1 m – at the top of the saltation layer at height $h_{\mathrm{salt}}$ – and the boundary condition for solving Equation (5) is given by relating the blowing snow mixing ratio $q_b$ at the lowest model level with the mixing ratio at the top of the saltation layer, $q_{\mathrm{salt}}$ (Eq. (1) in RACMO2 or Eq. (2) in the updated version) (Déry and Yau, 1999).

RACMO employs the double-moment PIEKTUK model. Therefore, an additional equation is employed to obtain the con-

centration of particles ($N$):

$$\frac{\partial N}{\partial t} = \frac{\partial}{\partial z}\left(K_N \frac{\partial N}{\partial z} + v_b N\right) + S_N. \tag{6}$$

Here, $K_N\ (\mathrm{m^2\,s^{-2}})$ is the eddy diffusivity for $N$, and $S_N\ (\mathrm{m^3\,s^{-1}})$ denotes the rate of change of particle numbers due to the sublimation process. The lower boundary condition for solving Equation (6) is here also the particle concentration at the top of the saltation layer ($N_{\mathrm{salt}}$) (Déry and Yau, 1999), which will be defined below.

In PIEKTUK-D model, the bulk blowing snow mixing ratio $q_b$ is related to $N$ via the spectral number density $F(r)$, following Schmidt (1982):

$$q_b = \frac{4\pi\rho_{ice}}{3\rho} \int_0^\infty r^3 F(r)\mathrm{d}r, \tag{7}$$

where the distribution of $F(r)$ follows two-parameter gamma distribution (Budd, 1966; Schmidt, 1982) by the relation:

$$F(r) = \frac{N r^{(\alpha-1)} \exp^{-r/\beta}}{\beta^\alpha \Gamma(\alpha)}, \tag{8}$$





where, $r$ represents the radius of ice particles, $\alpha$ (dimensionless) and $\beta$ (m) the shape and scale parameters of the gamma distribution $\Gamma$. Substituting, Equation (8) in (7), we obtain the particle number concentration $N_{\text{salt}}$ at the saltation layer:

$$N_{\text{salt}} = \frac{3\rho q_{\text{salt}}\Gamma(\alpha)}{4\pi\rho_{\text{ice}}\Gamma(\alpha+3)\beta^3},\qquad(9)$$

with $\alpha = 4.0$, $\beta = 100/\alpha$ ($\mu$m), and density of ice $\rho_{\text{ice}} = 917\ \text{kg}\,\text{m}^{-3}$. Equation (7) is discretised with the hypothesis that ice particle size follows two-parameter gamma distribution, with particle size bins covering particles of radius 2 to 300 $\mu$m (Déry

et al., 1998).

Finally, in the blowing snow model in RACMO, the mass change of an ice particle due to the blowing snow sublimation is given by the model of Thorpe and Mason (1966):

$$\frac{\mathrm{d}m}{\mathrm{d}t} = \left(2\pi r\sigma - \frac{Q_r}{KN_{\text{Nu}}T_a}\left[\frac{L_s}{R_v T_a}-1\right]\right)\bigg/\left(\frac{L_s}{KN_{\text{Nu}}T_a}\left[\frac{L_s}{R_v T_a}-1\right] + R_v\frac{T_a}{N_{\text{Sh}}De_i}\right),\qquad(10)$$

where $\sigma$ (dimensionless and negative) is the water vapour deficit with respect to ice $(e-e_i)/e_i$, where $e$ and $e_i$ are the
vapour pressure and its value at saturation over ice. $T_a$ is the ambient air temperature (K), $K$ the thermal conductivity of air ($\text{W}\,\text{m}^{-1}\,\text{K}^{-1}$), $L_s$ the latent heat of sublimation ($\text{J}\,\text{kg}^{-1}\,\text{K}^{-1}$), $R_v$ the gas constant for water vapour, and $D$ the molecular diffusivity of water vapour in air ($\text{m}^2\,\text{s}^{-1}$), $Q_r$ the net radiation transferred to the ice particle (W), and $N_{\text{Nu}}$ and $N_{\text{Sh}}$ being the Nusselt and Sherwood numbers.

### 2.3 Major changes to blowing snow model in RACMO

Six major updates in the implementation of PIEKTUK in RACMO are summarised below :

1. In RACMO, uniformly discretised 12-particle size bins were used, with a constant particle bin size $\Delta r = 4$ $\mu$m. Therefore, size bins with a mean particle radius greater than 50 $\mu$m were excluded, which caused the unexpected variation of $\text{TR}_{\text{ds}}$ observed in Fig. 1(b). Limited ice particle radius classes influenced the calculation of the bulk terminal velocity and the boundary conditions for Equations (6) and (5). To solve the issue, we use grid stretching technique similar to
DNS of channel flows to obtain non-uniform grids, with smooth stretching (Vinokur, 1983). We now use non-uniform discretisation with 16-particle size bins with varying $\Delta r$ to include all relevant particle size classes with particles of mean radius for each bin from 2 to 300 $\mu$m while keeping the computational overhead the same as before. Increase in the $\Delta r$ is non-uniform and follows a tangent hyperbolic function similar to the stretched grids used in DNS of channel flow . Déry et al. (1998) report convincing results by including particles of mean radius for each bin from 2 to 254 $\mu$m.

2. Previously, in the PIEKTUK model in RACMO, 32 vertical levels equidistant on a logarithmic scale were used. However, the vertical levels were not at the same height as the RACMO model levels, which made it difficult to include the blowing snow quantities as tendencies in the prognostic equations of water vapour and temperature. Furthermore, the blowing snow model was not fully coupled to the boundary-layer model in RACMO. Specifically, in the previous model version, instead of the simulated velocity profile from RACMO, log-law-based velocity and temperature profiles were
extrapolated from wind velocity and temperature at the first RACMO model level. In addition, the friction velocity ($u_*$)





was recalculated in the blowing-snow model, assuming near-neutral conditions. These inconsistencies have now been resolved, and actual velocity and temperature profiles and friction velocities from the RACMO boundary-layer model are used, which constitues another major improvement.

We have reduced the vertical levels to 16 to reduce computational expenses, with 8 logarithmically varying levels up to the lowest RACMO model level (dashed lines in Figure 2). The first model level is set to 0.1 m, which is assumed to be the top of the saltation layer ($h_{\text{salt}}$). Furthermore, above the lowest RACMO model level, the PIEKTUK-model levels coincide with the RACMO model levels, and this facilitates easier coupling of blowing snow sublimation as tendencies in the prognostic equations.

3. We found that the PIEKTUK model, when coupled with RACMO, is also highly sensitive to the model time step. While Déry and Yau (1999) specify a model time step of 2 seconds for PIEKTUK-B; in RACMO, the model time step was of the order of $300 - 600$ seconds. This time step was too large to predict the drift fluxes reliably. To overcome this, we introduce sub-stepping in the blowing snow model. By performing sensitivity analysis, we found that a constant time step of 10 seconds produces reliable estimates. Furthermore, the model quickly reaches a steady state in 5 sub-steps. Therefore, we use five sub-steps with a time step of 10 seconds and the fluxes from the last sub-step are taken as the representative flux for the whole RACMO model step.

4. In the original PIEKTUK model implementation by Déry and Yau (1999), the blowing snow mixing ratios are reset to zero only if the friction velocity is lower than the threshold friction velocity in two consecutive time steps, providing a realistic initial approximation of blowing snow quantities in each time step. However, previously in RACMO, $N$ and $q_b$ were reset to zero after every model time step, though in reality, the blowing snow events last for hours. Resetting the flux to zero is unrealistic and calls for a proper initialisation of the variables. Therefore, we now initialise $N$ and $q_b$ from the previous time step if two consecutive time steps satisfy the condition $u_* > u_{*t}$; otherwise, the values are reset to zero, indicating the end of the blowing snow event.

5. In RACMO, the bulk sublimation rate $S_b$ was used to calculate an integrated blowing snow sublimation flux, and this integrated moisture flux was added to the surface. While this approach works reasonably in obtaining SMB estimates, it is only partially physical since it limits the effect of blowing snow sublimation to the surface. To rectify this error in representation, we now add blowing snow sublimation rate $(-S_b)$ and latent heat due to blowing snow ($L_s S_b/c_p$) as tendencies to the prognostic equations of atmospheric water vapour and temperature, respectively.

For the PIEKTUK model levels below the lowest RACMO model level, a height-averaged tendency is calculated. It is used as the representative value for that RACMO level, and no moisture flux was added at the surface. For the rest of the RACMO model levels, the tendencies are obtained directly from the corresponding PIEKTUK vertical levels.

6. In RACMO, snowdrift was modelled if $u_* > u_{*t}$ and Equation (1) was used to estimate the saltation flux. This param- eterisation caused sharp variations in the saltation flux in RACMO and was not optimal. Therefore, the saltation flux is





now derived with Equation (2), which produces smooth variations of $q_{salt}$. Furthermore, Equation (2) is widely used in literature.

Finally, the formula to derive the vapour saturation pressure to ice ($e_i$) in Equation (10) has been updated to the AERKi formula (Alduchov and Eskridge, 1996; CY45R1—Part IV, 2018), as this formula is used in IFS code in which the blowing snow module is embedded.

## 3 Observation site and data

The observational site D47 (location: $67.4°$S, $138.7°$E, Fig. 1(a)), is located at an elevation of 1560 m, and at a distance of
105 km from the shore (Amory, 2020). Due to its topographical situation, the site experiences strong katabatic winds with a strong directional consistency. Due to the high surface winds, the site experiences frequent blowing snow events (Amory, 2020) and is ideally suited for evaluating RACMO results. For evaluation, observations of near-surface quantities such as 2-m wind speed, temperature, and air relative humidity are used, complemented with half-hourly drifting-snow transport fluxes. These observations are available for the years 2010–2012 with half-hourly temporal resolution. The drifting-snow transport fluxes
are measured with second-generation FlowCapt$^{TM}$ sensors. The sensors convert the acoustic vibration caused by blowing snow particles into integrated snow mass flux. The equipment consists of two 1-m length acoustic tubes, superimposed vertically to measure snow flux in the first 2 m above the ground. A detailed setup and observational site description can be found in Amory (2020) and Amory et al. (2020).

The blowing snow scheme in RACMO has multiple levels, with the lowest vertical level set at 0.1 m. For comparison with
observations, we obtain an average, vertically integrated, blowing snow flux, $Q_{T,RACMO}$ ($kg\,m^{-2}\,s^{-1}$), from the lowest model level upto 2-m height. Following Amory et al. (2021), since there are two acoustic tubes for measurement, we combine snow mass flux from both the tubes into an average, near-surface, mass flux $Q_{T,OBS}$ ($kg\,m^{-2}\,s^{-1}$), as :

$$Q_{T,OBS} = \frac{Q_{T,1}h_1 + Q_{T,2}h_2}{h_1 + h_2}, \tag{11}$$

where, $Q_{T,1}$ is the observed snow mass flux integrated over the exposed length of $h_i$ of the corresponding 2G-FlowCapt$^{TM}$
sensor. Furthermore, at site D47, wind speeds are available from the mast at a 2-meter height. Since the first atmospheric level in RACMO is above this height, we obtain the 2-m wind speed using the Monin-Obukhov similarity theory.

## 4 Results and discussion

Three RACMO simulations for 2000–2012, forced by ERA5 reanalysis data, were run for the evaluation presented here. The first one, hereon referred to as RpNew, employed all updates listed in Section 2.3. A second simulation was carried out with
the blowing snow scheme switched off, hereon referred to as NO-DRIFT, to study the effects of blowing snow compared to the no-blowing snow scenario. Finally, a simulation with the original blowing snow code of RACMO2.3p3, hereafter referred to as Rp3, has been carried out to compare the change in SMB estimates and related quantities. This simulation was needed as





the simulation presented by van Dalum et al. (2019) lacked the detailed blowing snow output needed for the study presented here.


## 4.1 Model evaluation with observations at site D47

### 4.1.1 Near-surface climate at site D47

**Table 1.** Root-mean-squared-error (RMSE), slope, intercept, bias, and coefficient of determination ($R^2$) of comparison of NO-DRIFT, Rp3, and RpNew simulations against observations at site D47. Statistics are reported for 2-m wind speed in $\mathrm{m\,s^{-1}}$, 2-m temperature in $^\circ\mathrm{C}$, 2-m relative humidity w.r.t ice in %, and the near surface blowing snow flux in $\mathrm{kg\,m^{-2}\,s^{-1}}$.

|  | NO-DRIFT | | | | Rp3 | | | | RpNew | | | |
|---|---|---|---|---|---|---|---|---|---|---|---|---|
|  | $U_{2m}$ | $T_{2m}$ | $RH_{2m}$ | $Q_T$ | $U_{2m}$ | $T_{2m}$ | $RH_{2m}$ | $Q_T$ | $U_{2m}$ | $T_{2m}$ | $RH_{2m}$ | $Q_T$ |
| Slope | 0.76 | 1.01 | 0.44 | - | 0.78 | 1.01 | 0.67 | 0.24 | 0.75 | 1.01 | 0.82 | 0.5 |
| RMSE | 3.84 | 3.09 | 18.84 | - | 3.69 | 3.17 | 9.39 | 0.04 | 3.88 | 3.04 | 6.64 | 0.03 |
| $R^2$ | 0.77 | 0.91 | 0.07 | - | 0.76 | 0.91 | 0.35 | 0.24 | 0.76 | 0.91 | 0.49 | 0.57 |
| Bias | -3.34 | 1.61 | -13.88 | - | -3.17 | 1.75 | -5.61 | 0.003 | -2.72 | 1.45 | -0.87 | -0.01 |

We evaluate the performance of RpNew in predicting the near-surface wind speed, temperature, relative humidity, and snow transport fluxes for 2010–2012 compared to the Rp3 and NO-DRIFT experiments. Site D47 is located in the katabatic wind
zone of the Antarctic coast and, therefore, receives high mean wind speeds (see Fig. 1(a)). Table 1 presents the statistics comparing observed near-surface quantities against simulated results from the three experiments. We observe that RACMO underpredicts the near-surface wind speed in all three experiments however with the current updates, the model bias is slightly decreased from -3.34 $\mathrm{m\,s^{-1}}$ in the NODRIFT case to -2.72 $\mathrm{m\,s^{-1}}$ in the case of RpNew. RACMO captures the variability in the data reasonably well, with negligible differences between the three experiments. The coefficient of determination ($R^2$) is
approximately 0.76, indicating that RACMO resembles the synoptic evolution of the wind strength well. A RMSE of approximately 3.88 $\mathrm{m\,s^{-1}}$ indicates that there are still significant differences between the model results and observations. As all three simulations underestimate the wind speed, we also performed tests with dual mass flux–TKE scheme (van Meijgaard et al., 2012), which allows better modelling of the turbulent boundary-layer processes. However, it did not improve the wind-speed predictions appreciably (not shown). Therefore, this scheme was not used further. The under-prediction of simulated wind
speed is likely due to the lower vertical resolution of the model, wherein the first atmospheric level is approximately 8 to 10 m above the surface, and the 2-m wind speed is calculated based on the similarity theory and is not simulated.

In the blowing snow model, the mass change of an ice particle due to the blowing snow sublimation is given by the model of Thorpe and Mason (1966) (Eq. (10)). Since the mass change depends on water vapour deficit and air temperature, accurate prediction of these quantities is necessary to obtain reliable estimates of blowing snow sublimation. Table 1 shows that the
near-surface temperature is overpredicted for all three experiments, and all simulations have a slight positive temperature bias.



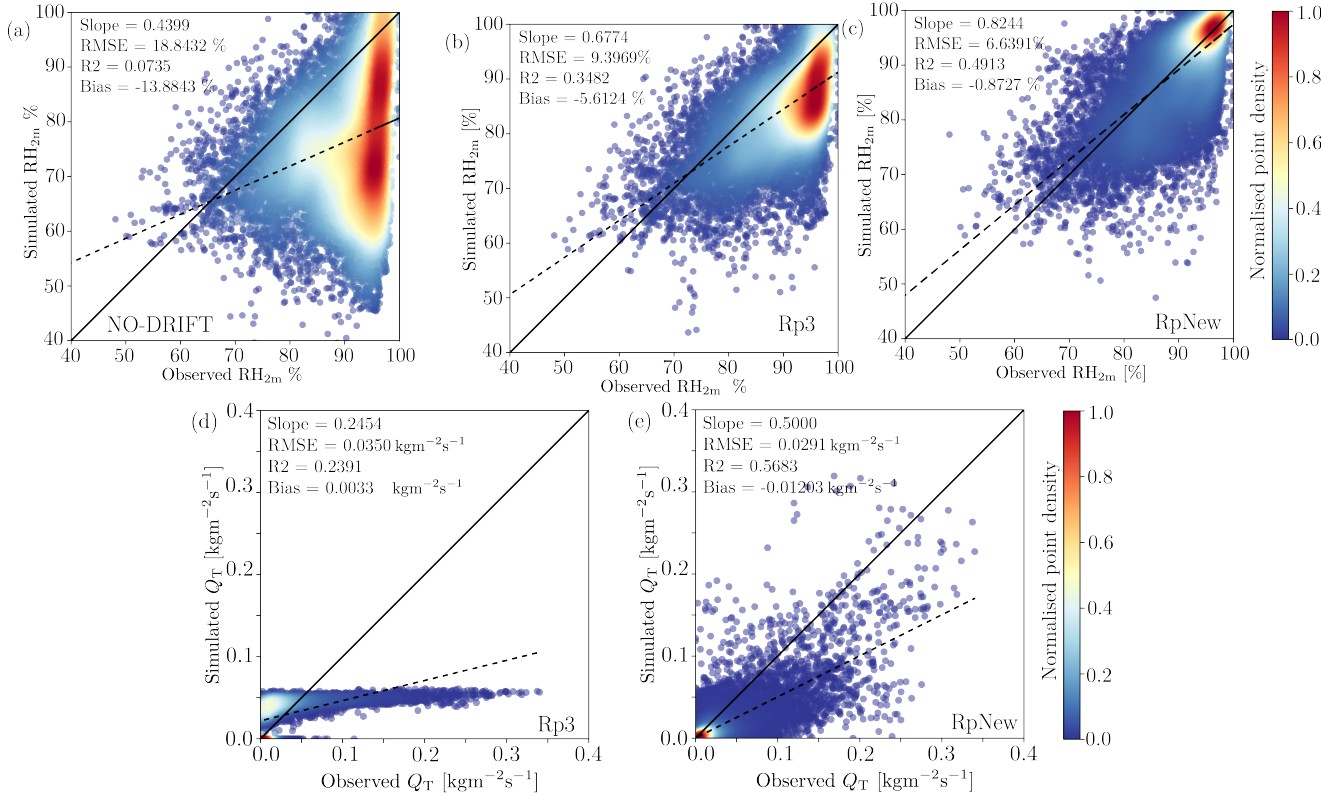

**Figure 3.** Density scatter plots of observed and simulated near-surface relative humidity w.r.t ice at site D47, for cases (a) NO-DRIFT, (b) Rp3, (c) RpNew, and Observed and simulated blowing snow fluxes $\mathrm{kg\,m^{-2}\,s^{-1}}$ (d) Rp3, and (e) RpNew. Solid lines represent the 1:1 line and the dashed lines represent the best-fit line. The colors represent the normalised point density from low (0.0, blue) to high (1.0, red).

However, with the updates to the model, the bias in the model is improved from 1.61 °C for the NO-DRIFT case to 1.45 °C for RpNew. RpNew also shows an improved temperature prediction with a lower bias of 0.3 °C compared to Rp3. The variability is modelled well, with an RMSE of 3°C, and a high $R^2 = 0.91$. The numbers show that RpNew predicts the near-surface wind and temperature better than the other two experiments.

Figure 3(a), (b), and (c) present a comparison of observed relative humidity with respect to ice against the simulated relative humidity for the three experiments. Figure 3(a) shows that the NO-DRIFT case shows a significant negative bias in the moisture, with low $R^2$ and a high error indicated by an RMSE of 18.84%. With Rp3, the results are slightly improved with a lower negative bias and a higher $R^2$, but still significant error in the simulated data. However, Figure 3(c) shows that with RpNew, the predictions are significantly improved compared to the other experiments. Though there is a significant spread in the data, the RMSE is 6.6%, and $R^2$ is 0.49, the figure shows a good match between the observed and simulated data. It is evident from Figure 3(a), (b), and (c) that the updates to RACMO significantly improve the moisture prediction when compared





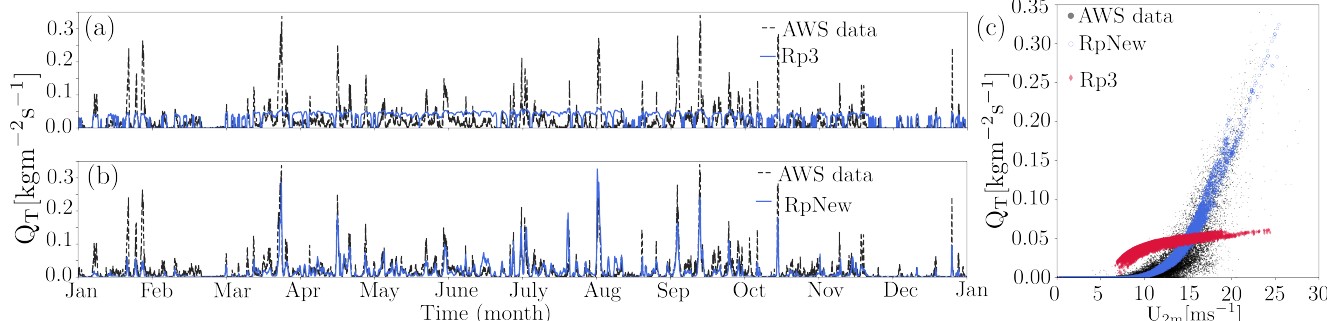

**Figure 4.** Comparison of simulated instantaneous near-surface blowing snow flux $Q_T$ [kgm$^{-2}$s$^{-1}$] with observations for the year 2011: (a) Rp3 (Rp3), (b) RpNew (c) Variation of near-surface blowing snow flux $Q_T$ with 2-m wind speed U$_{2m}$ [ms$^{-1}$]. Open circular, filled red diamond, and filled circular markers represent data from RpNew, Rp3, and AWS data, respectively.

with the observations. It shows that RACMO can produce reliable estimates of blowing snow sublimation despite uncertainties.

### 4.1.2 Blowing snow mass flux at site D47

In Figure 3(d) and (e), we present the comparison of simulated near-surface blowing snow mass flux with observed flux for Rp3 and RpNew, respectively. Simulated flux from Rp3 has a positive bias, with a very low $R^2$ indicating that Rp3 fails to capture the variability in the blowing snow flux observations. The predictions with Rp3 also have a higher RMSE of 0.035 kg m$^{-2}$s$^{-1}$. Also, it is apparent from Figure 3(d) that Rp3 fails to predict the blowing snow fluxes reliably when compared with the observations.

In contrast, with RpNew, we have a reasonable agreement between the observed and simulated fluxes (Fig. 3(e)) with R$^2$ = 0.56. The agreement indicates that the changes introduced in RACMO significantly improve its ability to predict the blowing snow fluxes reliably. Though the predictions are considerably improved compared to the observations, RACMO underpredicts the blowing snow fluxes. The underprediction is partially due to the under-prediction of velocities reported in table 1 and the model sensitivity to the chosen parameters. Since the snow transport flux varies in a power-law fashion with the

wind speed (Radok, 1977; Budd, 1966; Amory, 2020), the flux is highly sensitive to the wind-speed predictions; even a slight underprediction in the velocity introduces a significant difference in the blowing snow mass flux.

Figure 4(a) presents the instantaneous blowing snow mass flux obtained from Rp3 compared with the observations for the year 2011. It is evident from the figure that Rp3 does not reliably predict the blowing snow magnitude. In RACMO2.3p2 and Rp3, the linear saltation coefficient $c_{salt}$ (Eq. (1)) was reduced (van Wessem et al., 2018) which resulted in the low, capped snow

transport flux seen in Figure 4(a). As halving $c_{salt}$ roughly led to halving the snow drift flux, the RACMO versions preceding version 2.3p2, with doubled $c_{salt}$, overestimated $Q_T$ for most of the time (not shown).

Figure 4(b) presents the instantaneous blowing snow mass flux obtained with the RpNew for the year 2011. It is evident from the figure that the RpNew works reasonably well in predicting both the magnitude and occurrence of the blowing snow.





Specifically, the flux matches the observations reasonably well in the Antarctic winter (March–October). However, it is under-
predicted in the Antarctic summer (October–March). Underprediction might be related to the amount of loose snow available at
the surface, possibly due to inaccuracies of the modelled surface snow compaction in summer or the direct interaction between
precipitation and snow drift, which RACMO does not resolve. As we found no clear cause for the underestimation of snow
drift during summer, further study is necessary to uncover the seasonal differences in the blowing snow flux. From Figures 4(a)
and (b), it is evident that RpNew successfully predicts the blowing snow mass flux reliably, in contrast to Rp3, and can be used
to obtain reliable continent-wide estimates of sublimation in the polar regions.

Figure 4(c) presents the variation of blowing snow mass flux with the near-surface wind speed. As mentioned previously,
blowing snow mass flux is expected to vary in a power-law fashion with the wind speed. Clearly, flux from Rp3 fails to produce
this behaviour; however, RpNew successfully predicts the power-law variation of the blowing snow mass flux. The primary
reason for this improvement is the non-uniform ice particle radius distribution, allowing us to include all relevant ice particles
in the range between 2 to 300 $\mu$m. Coupled with the better coupling with RACMO prognostic variables and sub-stepping, the
behaviour of the flux follows the expected power-law variation seen in Figure 4(c).

Though RpNew results show the desired behaviour, it fails to capture the spread in the observational data. Through sensi-
tivity analysis of the data, we found that the spread in the data depends on the modelling choices made, e.g. the parameter $\alpha$
in two-parameter gamma distribution (equation 8) and the threshold friction velocity. Budd (1966) and Schmidt (1982) report
that the distribution of ice particle diameters follows a two-parameter gamma function that varies with height from the ground,
with $\alpha$ value varying between 2 and 14. However, for simplified implementation, following Déry and Yau (2002), we used a
constant $\alpha = 4$, which does not vary with height; this influences the modelled snow mass flux at different heights. Furthermore,
we use constant snow grain properties with dendricity $d = 0.5$ and sphericity $s = 0.5$ in the calculation of snow mobility index
MO (Lenaerts et al., 2012); these snow grain properties influence the calculation of the threshold friction velocity (Eq. (3))
which can cause different snow flux at same velocities, influencing the spread in the data. These simplifications inherent in the
blowing snow model affect the model results; regardless of simplifications, it is evident from the results that RpNew success-
fully predicts blowing snow fluxes with reasonable accuracy.

### 4.1.3 Blowing snow events at site D47

To quantify the ability of RACMO to predict a blowing snow event accurately, we follow Amory et al. (2017, 2021) and clas-
sify blowing snow events as the occurrences when the blowing snow mass flux is greater than $10^{-3}\,\mathrm{kg\,m^{-2}\,s^{-1}}$. Subsequently,
we create confusion matrices comparing the blowing snow events from observed and simulated data. The diagonal entries in
the confusion matrix represent the blowing snow events correctly predicted by the simulations, and the off-diagonal entries
represent the events not correctly predicted by the simulations.

Table 2(a) and (b) represent the confusion matrices that provide the percentage of blowing snow events observed and sim-
ulated by the RpNew and Rp3, respectively. In table 2(a), we see that out of the total observations, there are 80% of observed
blowing snow events, and RpNew manages to predict 54% of these blowing snow events. In contrast, Rp3 manages to predict




**Table 2.** Confusion matrix presenting the comparison between observed and simulated blowing snow events. DRIFT represents the events where $Q_T > 10^{-3}$ [kg m$^{-2}$ s$^{-1}$] and NO-DRIFT represents the remaining events.

(a) RpNew

| OBS \ SIM | NO-DRIFT | DRIFT |
|---|---|---|
| NO-DRIFT | 18% | 2% |
| DRIFT | 26% | 54% |

(b) Rp3

| OBS \ SIM | NO-DRIFT | DRIFT |
|---|---|---|
| NO-DRIFT | 16% | 4% |
| DRIFT | 17% | 63% |

**Table 3.** Confusion matrix presenting the high-mass flux blowing snow events. Diagonal elements represent the events that are correctly classified between observations and simulations.

(a) RpNew

| OBS \ SIM | $Q_T \leq 0.05$ | $Q_T > 0.05$ |
|---|---|---|
| $Q_T \leq 0.05$ | 84% | 2% |
| $Q_T > 0.05$ | 9% | 5% |

(b) Rp3

| OBS \ SIM | $Q_T \leq 0.05$ | $Q_T > 0.05$ |
|---|---|---|
| $Q_T \leq 0.05$ | 86% | 0% |
| $Q_T > 0.05$ | 14% | 0% |

63% of the total blowing snow events. We calculate the blowing snow frequency as the ratio of correctly simulated blowing snow events and the total number of observed blowing snow events. For RpNew, we obtain a blowing snow frequency of 0.68, and Rp3 has a blowing snow frequency of 0.79. Clearly, Rp3 manages to identify most of the blowing snow events, however Fig. 4(a) shows that it does not capture any peaks in the blowing snow fluxes.

To evaluate the performance of RpNew in identifying the higher magnitude blowing snow fluxes, we create another confusion matrix where we compare the blowing snow events with blowing snow mass flux $Q_T > 0.05$ kg m$^{-2}$ s$^{-1}$. Table 3(a) and (b) present a comparison of the observed and simulated blowing snow fluxes for events with $Q_T > 0.05$ kg m$^{-2}$ s$^{-1}$. The tables show that 14% of the observed events account for events with high blowing snow mass flux. While RpNew captures 5% out of the 14% high blowing snow events, Rp3 does not capture any of these events. Specifically, RpNew successfully predicts 36% of the observed events with a high blowing snow mass flux, a marked improvement compared to Rp3. This underprediction of strong snow drift events is closely related to the underestimation of the wind speed (Table 1), as there is no apparent underestimation in the modelled strength of snow drift events when plotted as a function of the near-surface wind speed (Fig. 4c). The results show that RpNew provides reasonable estimates of low- and high-magnitude blowing snow events while future improvements are needed.

### 4.1.4 Blowing snow sublimation at site D47

Figure 5(a) shows the modelled instantaneous profiles of blowing snow sublimation rate for 2011 at site D47. In the winter, deep blowing snow layers are modelled, with a typical range of blowing snow layer heights and snow sublimation between 100





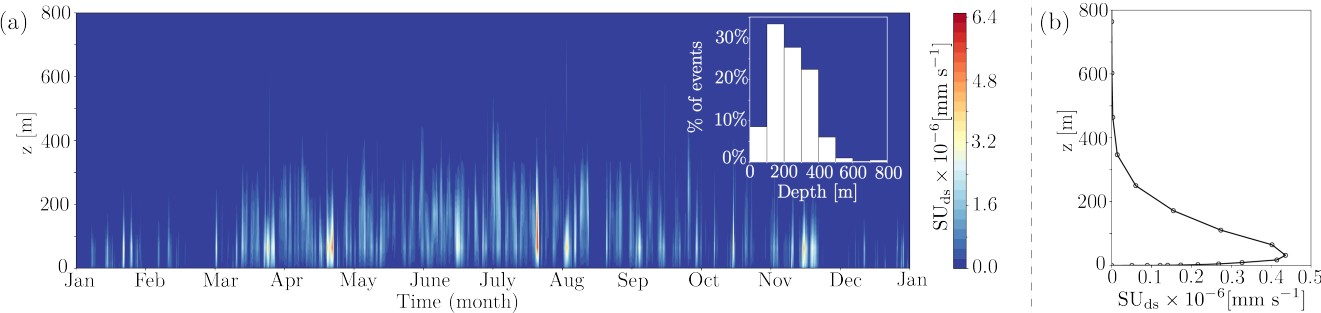

**Figure 5.** Yearly mean (2011): (a) Yearly average blowing snow sublimation rate for the year 2011 in $\mathrm{mm\,s^{-1}}$, inset shows the histogram of blowing snow layer depth (m). (b) Variation of blowing snow sublimation rate with height ($\mathrm{mm\,s^{-1}}$).

and 500 m. In summer, a shallower blowing snow layer is modelled. The Figure shows multiple events with continuous blowing snow storms in winter, indicating a significant contribution of blowing snow to Antarctic sublimation. Although sublimation over a thick layer coincides with blowing snow events (Fig. 4b), we do not see a direct relation between the near-surface snow drift flux and the intensity or total magnitude of blowing snow sublimation. This shows the necessity to explicitly couple the blowing snow model to the atmospheric model layers, as the modelled temperatures, humidities, and wind speeds of the

lowermost model level are unlikely representative of the whole boundary layer.

Figure 5(b) presents the yearly averaged blowing snow sublimation rate profile for the year 2011 at site D47. The average blowing snow layer depth is $230 \pm 116$ m. As the air is saturated at the surface, the sublimation at the surface is negligible, with sublimation increasing away from the ground and maximum sublimation above. Déry and Yau (2002) and Toumelin et al. (2021) have reported a similar variation of the blowing snow sublimation. It is worth noting here that both the drifting snow

concentration and horizontally drifting snow transport are peaking even close to the ground (not shown). As depicted in Figure 5(a), blowing snow sublimation starts well below the first RACMO model level as the lowermost PIEKTUK model layers are saturated with respect to water vapour. Blowing snow sublimation thus occurs in the upper part of the boundary layer, where the blowing snow concentration is low, but the air is not yet saturated.

Based on lidar data from CALIPSO (Cloud-Aerosol Lidar and Infrared Pathfinder Satellite Observation), Palm et al. (2017)

report for the Antarctic Ice Sheet north of 82 °S an average snow layer depth of 120 m, with typical blowing snow layers of 200 m all along the coastal katabatic wind regions (see Fig. 5 in Palm et al. (2018)). For the site D47, RpNew shows a similar mean layer depth of $230 \pm 116$ m, and a similar typical range (Inset Fig. 5(a)).This analysis shows that RpNew satisfactorily reproduces all the necessary features of the blowing snow sublimation and can be used to obtain continent-wide estimates. However, it is worth mentioning that total blowing snow sublimation is sensitive to horizontal resolution. At the 27

km resolution employed in the study, strong spatial gradients near the coast would not be accurately captured. Subsequently, the impact of blowing snow on sublimation and horizontal transport of mass can be underestimated.




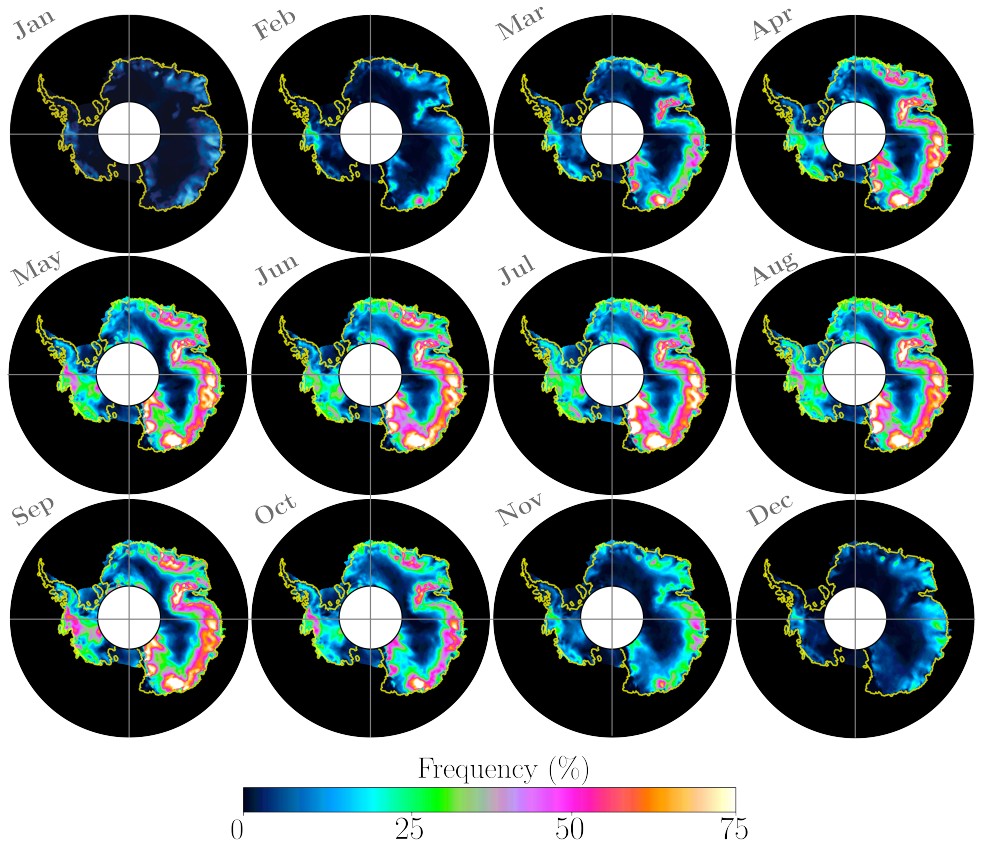

**Figure 6.** Blowing snow frequency visualised to provide a qualitative comparison with satellite measurements of Palm et al. (2018). Figures show average blowing snow frequency over the decade 2001–2010.

## 4.2 Continental blowing snow frequency

Figure 6 gives the monthly variation of mean blowing snow frequency over Antarctica for the decade 2001–2010. Blowing snow frequency is obtained by calculating all the blowing snow events with the blowing snow mixing ratio $q_b > 10^{-6} \, \mathrm{kg \, kg^{-1}}$.

The cutoff $q_b$, the limits, and the colourmap in Figure 6 are chosen to facilitate a qualitative comparison with the satellite observations presented in Figure 3 in Palm et al. (2018). We observe that the monthly blowing snow frequency largely follows the seasonal trend in the surface wind patterns over Antarctica, with high-frequency blowing snow in winter compared to summer. Whereas Figure 4(b) suggests that RpNew underestimate summer snow drift, such summer underestimation is not very apparent between RACMO and the satellite observations. It is worth mentioning here that the blowing snow frequency

presented in Figure 6 includes near-surface blowing snow flux. In contrast, the satellite observations by Palm et al. (2018) include only those blowing snow layers deeper than 30 m and only those events without clouds. Despite the differences, the simulated blowing snow frequency is qualitatively similar to that obtained from the CALIPSO satellite observations (Palm et al., 2017, 2018). The results show a persistent blowing snow hotspot in East Antarctica near Adélie Land, observed in



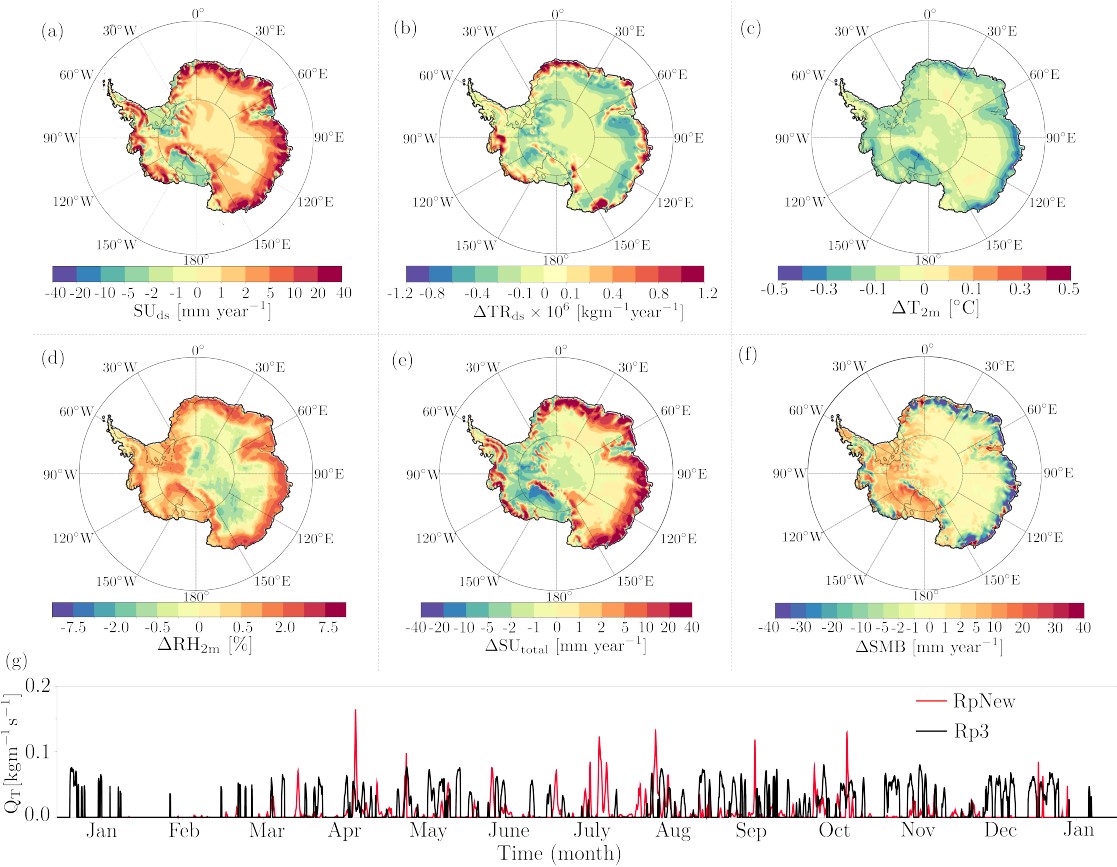

**Figure 7.** Yearly averaged (2000-2010) difference between RpNew and Rp3 quantities: (a) Blowing snow sublimation [mm yr$^{-1}$], (b) Blowing snow transport [kg m$^{-1}$yr$^{-1}$], (c) Near-surface temperature $T_{2m}$ in $^{\circ}C$, (d) Relative humidity in percentage, (e) Total sublimation [mm yr$^{-1}$], and (f) SMB [mm yr$^{-1}$]. $SU_{ds}$, $SU_{total}$, and SMB are in mm water equivalent. (g) Instantaneous drifting snow flux at an interior region of East Antarctica (71.1$^{\circ}$S, 111.7$^{\circ}$E).

satellite observations and our simulations. We can also infer that the satellite observations slightly underpredict the frequencies
compared with the simulations for the reasons above.

Our results are qualitatively similar to the simulations with the CRYOWRF model carried out by Gerber et al. (2023). Though there are differences between the studies, the results are qualitatively comparable. Most of the blowing snow hotspots observed in our simulations also correspond to the 'wind glaze' areas in East Antarctica reported by Scambos et al. (2012). Scarchilli et al. (2010) report blowing snow frequencies of 80% at the wind convergence zone of Terra Nova Bay (East Antarctica); we
observe approximately 80–90% blowing snow frequency in the area during the winter months.





## 4.3 Difference between RpNew and Rp3

In Figure 7, we present the difference in some important variables between RpNew and Rp3 to quantify the magnitude of change between the two versions. The blowing snow transport $TR_{ds}$ (Fig. 7(b)) decreased somewhat over most of Antarctica with significant but localised increases in transport along George V Land, Adélie Land, and Dronning Maud Land. At these

locations, the blowing snow transport is increased by 2 - 3 times compared to Rp3 due to better modelling of snow particle distribution, which includes more particles with well-distributed ice particle radii and particle initialisation. As visualised in Figure 7(g) as an example, however, for most of Antarctica, most blowing snow events are reduced in intensity by the model updates. Only a few instances per year does the wind speed exceed the threshold for which the updated blowing snow model simulates higher blowing snow transport.

Conversely, for most of Antarctica, we observe higher blowing snow sublimation (Fig. 7(a)) due to the ability of RpNew to capture the peaks in blowing snow fluxes and the change in initialisation employed for the blowing snow model. This increase indicates the necessity of a direct two-way coupling of the modelled atmospheric profile of the surface layer to the blowing snow model. In the RpNew, the snow particles are lifted into the warmer and drier air of the upper part of the stable boundary layer. In Rp3, particles were not lifted that high - due to errors in the particle size distribution - not seeing this warmer air due

to using an extrapolated 10 m temperature profile assuming neutral conditions. The larger ice shelves are the only regions of Antarctica where blowing snow sublimation decreases. Here, the stable boundary layer is generally very thick (e.g. van den Broeke and Van Lipzig (2003), Fig. 10), inhibiting the blowing snow from reaching the warmer air above the surface layer.

Results show RpNew is slightly colder by 0.3–0.4 K (Fig. 7(c)) along the coastal areas when compared to Rp3. This results from a better coupling of blowing snow sublimation to the tendencies of temperature, which allows the removal of latent heat

from upper vertical levels of RACMO. Compared to Rp3, RpNew has higher relative humidity (Fig. 7(d)) also, due to better coupling of blowing snow moisture tendencies with RACMO, the change in moisture leads to an increase in the dew-point temperature of $2-4$ K (not shown here) in the first few vertical layers of RACMO. Furthermore, the total sublimation is higher in RpNew (Fig. 7(e)) when compared to Rp3. Along the coast, the difference is as high as $100\ \mathrm{mm\,w.e.\,yr^{-1}}$. Overall, the averaged surface mass balance (Fig. 7(f)) is changed mostly along the coastal Antarctica with a reduction of approximately 30

$-40\ \mathrm{mm\,w.e.\,yr^{-1}}$. Since there is an increase in the moisture availability, there is relatively higher precipitation over Ronne and Ross ice shelves with a corresponding increase in SMB of approximately $20\ \mathrm{mm\,w.e.\,yr^{-1}}$.

In conclusion, changes introduced in RpNew greatly influence the overall sublimation pattern in Antarctica and moisture content in lower levels of the atmosphere. In the RpNew, blowing snow's impact is more regional than Rp3. However, the overall impact on SMB is limited, with a decrease in SMB on the Eastern Antarctic coast and a slight increase in SMB in Western

Antarctica due to higher moisture content created by blowing snow sublimation.



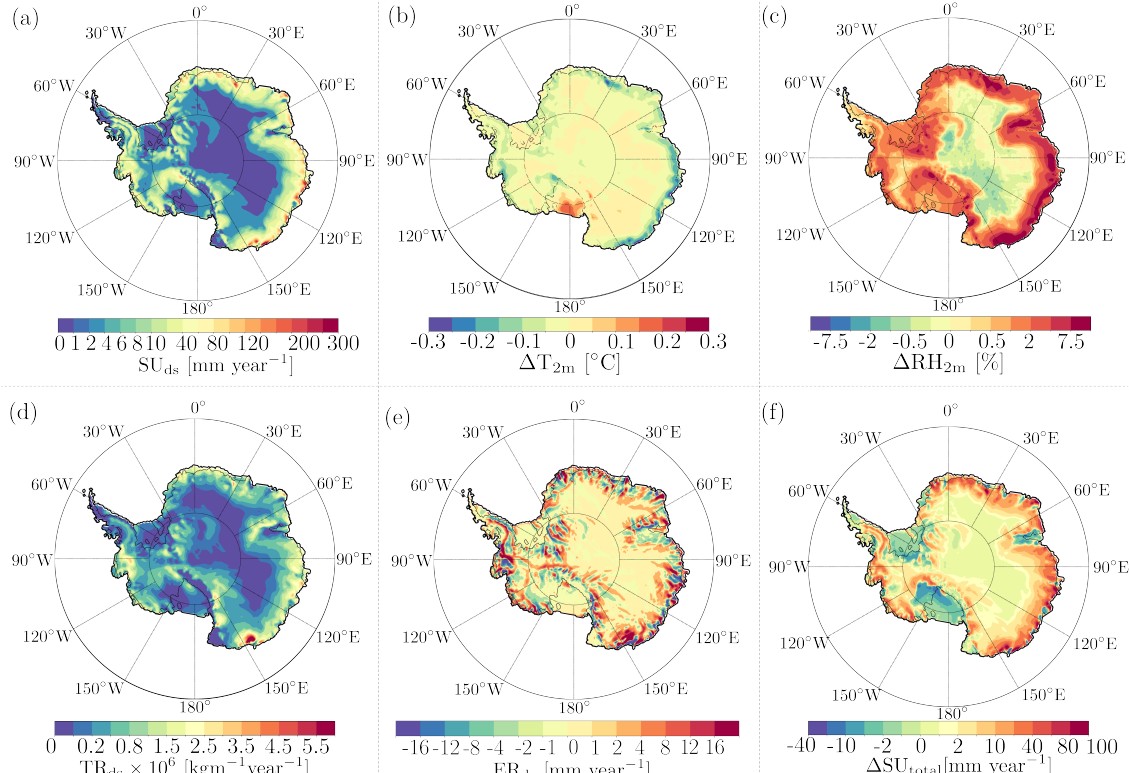

**Figure 8.** Yearly mean (2000-2010): (a) Blowing snow sublimation $SU_{ds}$ in mm water equivalent, (b) Difference in near-surface temperature, $\Delta T_{2m}$ between RpNew and NO-DRIFT simulations, (c) Difference in relative humidity $\Delta RH_{2m}$, (d) Blowing snow flux [$kgm^{-1}yr^{-1}$], (e) Erosion due to blowing snow, $ER_{ds}$ or the divergence due to blowing snow [$kgm^{-1}yr^{-1}$], and (f) Total sublimation $SU_{total}$, including surface- and blowing-snow sublimation [$kgm^{-1}yr^{-1}$].

## 4.4 Continent-wide estimates of blowing snow climate over Antarctica

Figure 8 presents the updated continent-wide estimates of the blowing snow climate of Antarctica by comparing the yearly average (2000-2010) quantities of RpNew and NO-DRIFT simulations.

Similar to previous model results (Lenaerts and van den Broeke (2012)), we observe negligible blowing snow sublimation (Figure 8(a)) in the interior parts of Antarctica with maximum sublimation towards the coast. Model results show blowing snow sublimation hotspots ($SU_{ds} > 100\ mm\,w.e.\,yr^{-1}$) in George V Land, Adélie Land, Wilkes Land, and Queen Mary Land in Eastern Antarctica with non-zero sublimation all along the coast of Antarctica. RACMO shows negligible sublimation over Dome Fuji, Dome Argus, and Dome C, which form the interior parts of Eastern Antarctica, due to the lower wind speed and low temperatures in these regions (Fig. 1(a)). Similarly, we observe negligible sublimation over Ronne and Ross ice shelves,
which also experience low wind speeds. This shows that blowing snow sublimation is mostly limited to the katabatic wind regions of Antarctica. Maximum blowing snow sublimation of $335 \pm 30\ mm\,w.e.\,yr^{-1}$ occurs in Adélie Land at the location 66.9°S, 130.4°E. Palm et al. (2017) based on the CALIPSO lidar observations report a maximum blowing snow sublimation





of $250 \pm 125 \,\mathrm{mm\,w.e.\,yr^{-1}}$ near the coast between longitudes 140 and 150°E. Both spatial distribution and the magnitude of

blowing snow sublimation from RpNew match reasonably well with CALIPSO observations of Palm et al. (2017).

Figure 8(b) provides the difference between the 2-meter temperature for the RpNew and NO-DRIFT cases. The figure shows that blowing snow sublimation reduces the near-surface temperature. At blowing snow sublimation hotspots, we observe a cooling of $0.1 - 0.3 \,\mathrm{K}$, with negligible change in the temperature over most of interior Antarctica. It is worth mentioning here that with Rp3, we observed a 'warm' bias compared to the case with NO-DRIFT (not shown); this shows that the coupling was

incorrect in the previous version of RACMO. The results have appreciably improved with RpNew. However, the overall effect of blowing snow sublimation on the yearly average near-surface temperature in Antarctica seems negligible similar to previous model results.

Higher sublimation due to blowing snow in RpNew to lead to higher near-surface relative humidity (Fig. 8(c)) when compared to NO-DRIFT simulations. We observe higher relative humidity along the Antarctic coast with a maximum of 10% in the

coastal George V Land and Adélie Land. This increase in relative humidity is higher when compared to what was previously observed with RACMO. Similar to sublimation, blowing snow transport $\mathrm{TR_{ds}}$ $(\mathrm{kg\,m^{-1}\,yr^{-1}})$ (Fig. 8(d)) is negligible over interior Antarctica. We observe a strong blowing snow transport near coastal George V Land with maximum transport of $9\times10^6$ $\mathrm{kg\,m^{-1}\,yr^{-1}}$. Along the rest of the Antarctic coast, blowing snow transport is approximately $2 \times10^6 - 3 \times10^6 \,\mathrm{kg\,m^{-1}\,yr^{-1}}$. Blowing snow erosion $\mathrm{ER_{ds}}$ $(\mathrm{mm\,w.e.\,yr^{-1}})$ (Fig. 8(e)) which is a contributor to Antarctic SMB, shows complex convergence

and divergence patterns all along the Antarctic coast. Similar to Bromwich et al. (2004) and Lenaerts and van den Broeke (2012), we observe large blowing snow divergence near escarpment areas with significant katabatic wind acceleration. Furthermore, areas with blowing snow convergence are near blowing snow divergence, which indicates that blowing snow is important for redistributing the precipitation in the coastal areas of Antarctica. However, the magnitude of $\mathrm{ER_{ds}}$ is not significant enough for a major contribution to SMB, as only the snow blown off Antarctica counts for the integrated SMB.

Total sublimation $\mathrm{SU_{total}}$ $(\mathrm{mm\,w.e.\,yr^{-1}})$, sum of blowing snow and surface sublimation $(\mathrm{SU_{ds} + SU_s})$ follows the spatial distribution of blowing snow sublimation. Maximum total sublimation of $396 \,\mathrm{mm\,w.e.\,yr^{-1}}$ is observed at the same location as the maximum blowing snow sublimation indicating the leading contribution of blowing snow sublimation to the total sublimation. Total sublimation is higher in the RpNew simulations when compared to NO-DRIFT simulations (Figure 8(f)); in the regions near Adélie Land, the difference in total sublimation is as high as $200 \,\mathrm{mm\,w.e.\,yr^{-1}}$. In the absence of blowing

snow sublimation, total sublimation is under-predicted, and therefore, blowing snow sublimation should be included in the calculations of SMB.

## 4.5 Seasonal variation of integrated sublimation

Monthly contribution to the yearly average integrated blowing snow sublimation from Rp3 and RpNew (Fig. 9(a)) shows that the blowing snow sublimation is lower in summer compared to winter. Lower sublimation is due to higher temperatures and

summer snow densities, making it difficult for the snow to lift off from the ground. Blowing snow sublimation $\mathrm{SU_{ds}}$ increases with the onset of winter and remains relatively constant over winter with an approximate contribution of $15 - 20 \,\mathrm{Gt\,mo^{-1}}$ in winter. Constant blowing snow indicates that blowing snow sublimation is a major contributor to total sublimation in winter.





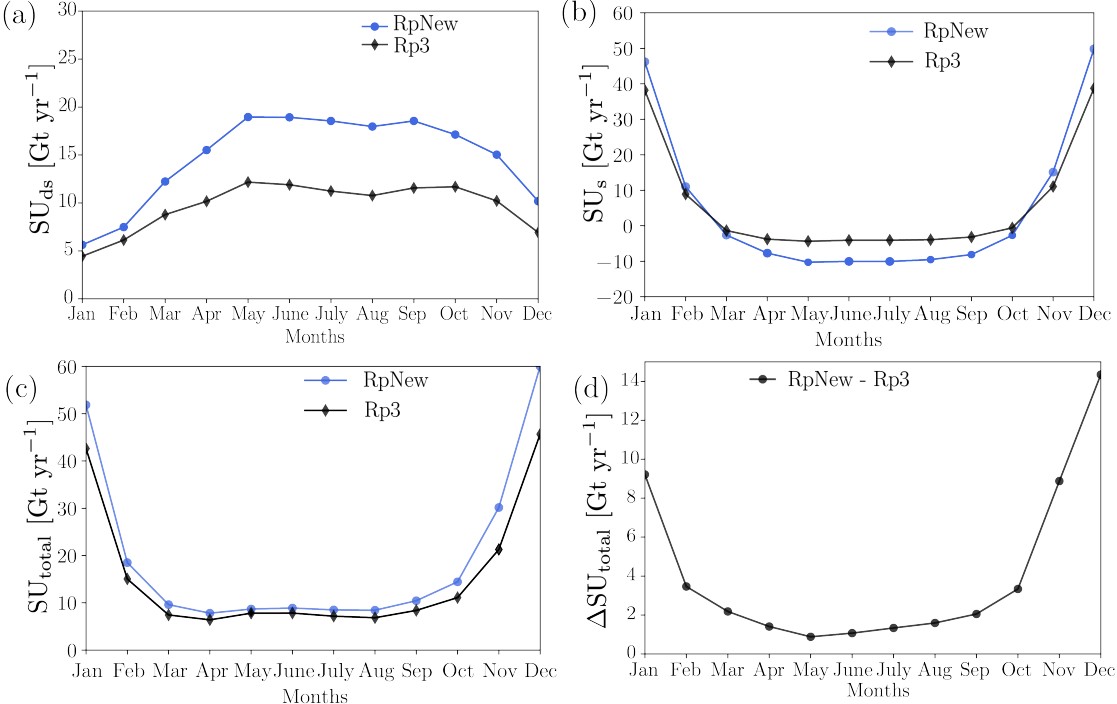

**Figure 9.** Monthly contribution to the yearly mean (2000–2010) (all in $\mathrm{Gt\,mo^{-1}}$) (a) integrated blowing snow sublimation over total ice sheet, (b) integrated surface sublimation, and (c) integrated total sublimation (SU$_{\mathrm{tot}}$ = SU$_{\mathrm{ds}}$ + SU$_{\mathrm{s}}$), (d) the difference between integrated SU$_{\mathrm{total}}$ in the RpNew and NO-DRIFT.

Compared to Rp3, the blowing snow in RpNew is nearly doubled all through the winter. The reason for the doubling of the blowing snow sublimation is multi-fold. Firstly, the model coupling with the prognostic equations of temperature and humid-
ity is better (improved physics); specifically in RACMO, the coupling was done assuming a neutral boundary layer and the velocities were assumed as logarithmically varying without passing the actual RACMO velocities to the blowing snow model. Therefore, the blowing snow model has a better overall sense of the atmosphere above the first RACMO model level and senses that the temperatures are higher and the air is dryer above. Secondly, the blowing snow model has a better initialisation and sub-stepping. Specifically, in the previous RACMO version, the blowing snow model was reset to zero after every time step. At
the same time, this is true; in the absence of sub-stepping, the blowing snow particle concentration never reaches the equilibrium, resulting in under-prediction blowing snow particle concentration. With updated numerics and better model initialisation, RpNew has improved physics, resulting in large differences in the blowing snow sublimation between Rp3 and RpNew.

Surface sublimation SU$_{\mathrm{s}}$ dominates the sublimation in summer due to higher temperatures (Fig. 9(b)) and reaches a relatively constant value in winter. In winter, between March and November, we observe negative surface sublimation, hence the de-
position of water vapour onto the snow surface. This deposition agrees with the measurements of King et al. (1996), who measured small, downward water vapour fluxes in the winter of 1991 at Halley station, East Antarctica. A similar seasonal





cycle in the surface sublimation with a negative surface sublimation in winter has been reported by King et al. (2001). While the blowing snow sublimation is increased in RpNew, the negative surface sublimation also increases, balancing the net change in total sublimation. The deposition follows the same spatial and seasonal pattern as the blowing snow sublimation. Since the
condensation is directly proportional to the difference between vapour pressures of water at the surface and above the surface, with RpNew, which has better coupling with the atmosphere, there is more condensation in winter compared to Rp3.

Total sublimation $SU_{total}$ (Fig. 9(c)) follows a similar pattern as the surface sublimation, with higher values during summer and relatively constant values in winter. Figure 9(d) presents the difference between total sublimation and surface sublimation, which shows the difference is more or less equal to blowing snow sublimation. The difference in the total sublimation between
Rp3 and RpNew shows that the major sublimation changes are observed in summer, with higher sublimation in summer with RpNew. While the seasonal trends of the sublimation remain unaltered between Rp3 and RpNew, interestingly, we observe that an increase in blowing snow sublimation in winter leads to an increase in deposition, leading to marginal overall changes to total sublimation. However, as Figure 7(e) shows, these limited changes in the spatially mean sublimation result from rather large but opposing regional changes in total sublimation.

## 4.6 Changes in integrated SMB

**Table 4.** Total ice sheet, including ice shelves, integrated SMB mean 2000–2010 values ($\mathrm{Gt\,yr^{-1}}$) with interannual variability $\sigma$: total (snow and rain) precipitation ($P_{tot}$), total sublimation ($SU_{tot}$), surface sublimation ($SU_s$), blowing snow sublimation ($SU_{ds}$), blowing snow erosion ($ER_{ds}$), run-off (RU). Integrated surface mass balance is given by: $SMB = P_{tot} - SU_{ds} - SU_s - ER_{ds} - RU$. (a) Difference between RpNew and RP3, and (b) SMB difference between RpNew (2000-2010) with CRYOWRF (2010–2020) (Gerber et al., 2023)

|  | (a) RpNew and Rp3 |  |  |  |  |  | (b) RpNew (2000–2010) and CRYOWRF (2010–2020) |  |  |  |  |
|---|---|---|---|---|---|---|---|---|---|---|---|
|  | RpNew |  | Rp3 |  | Difference |  | RpNew |  | CRYOWRF |  | Difference |
|  | mean | $\sigma$ | mean | $\sigma$ | mean |  | mean | $\sigma$ | mean | $\sigma$ | mean |
| $P_{tot}$ | 2696 | 97 | 2674 | 99 | +22(0.8%) | $P_{tot}$ | 2696 | 97 | 3101 | - | −405(−14%) |
| $SU_{tot}$ | 237 | 10 | 188 | 5 | +49(23%) | $SU_{tot}$ | 237 | 10 | 335 | - | −98(−34%) |
| $SU_s$ | 61 | 8 | 72 | 5 | −11(−16%) | $SU_s$ | 61 | 8 | 234 | - | −173(−117%) |
| $SU_{ds}$ | 176 | 7 | 116 | 4 | +60(41%) | $SU_{ds}$ | 176 | 7 | 101 | - | +75(54%) |
| $ER_{ds}$ | 8 | 0.5 | 5 | 0.2 | +3(46%) | $ER_{ds}$ | 8 | 0.5 | 31 | - | −23(−118%) |
| RU | 7 | 3 | 7 | 3 | 0 | RU | 7 | 3 | 5 | - | +2 (+33%) |
| SMB | 2444 | 100 | 2474 | 99 | −30(1.2%) | SMB | 2444 | 100 | 2730 | - | −286(−11%) |

Table 4(a) presents the SMB and its components integrated over the whole ice sheet (including ice shelves) for the years 2000 – 2010 in $\mathrm{Gt\,yr^{-1}}$ along with their inter-annual variability. Compared to Rp3, RpNew has an increased precipitation of 22 $\mathrm{Gt\,yr^{-1}}$ caused by the higher moisture content in the atmosphere due to higher blowing snow sublimation. The total subli- mation is increased by 49 $\mathrm{Gt\,yr^{-1}}$ with blowing snow sublimation being the major contributor. There is a slight decrease in





surface sublimation ($11 \, \mathrm{Gt\,yr^{-1}}$) as air in the boundary layer is saturated more efficiently with RpNew compared to Rp3, which causes a reduction in the potential for the surface sublimation. With higher blowing snow transport fluxes, we have a higher snow erosion increase of $3 \, \mathrm{Gt\,yr^{-1}}$. This number remained small as snow erosion only influences the integrated SMB once the snow is blown off the ice sheet. Overall, the integrated SMB is reduced by $30 \, \mathrm{Gt\,yr^{-1}}$, due to a net increase in blowing snow sublimation. The change amounts to only a 1.2% decrease in SMB compared to Rp3. Since the change in SMB with the updates is minor and the SMB results from RACMO have been previously evaluated against several in-situ and remote sensing observations, we refer to Noël et al. (2018); van Wessem et al. (2018) for the SMB evaluation. Though there is negligible change in the overall SMB, blowing snow sublimation is highly important to local SMB, especially in the escarpment areas in Eastern Antarctica.

Recently Gerber et al. (2023) carried out simulations of Antarctic climate at 27 km resolution using the CRYOWRF model. Table 4(b) compares the integrated quantities obtained from RpNew with CRYOWRF. It is worth noting here that the experiments with CRYOWRF were carried out from 2010 to 2020, while our results are for 2000 to 2010. While the time period is different, Gerber et al. (2023) is the only other study (other than RACMO studies) that reports SMB results of the entire Antarctica with a blowing snow model, making these results interesting to compare. Table 4(b) shows that there is a large difference in SMB (11%) and precipitation (14%) between RpNew and CRYOWRF. While precipitation and SMB are comparably higher in CRYOWRF, the ablation terms of CRYOWRF, especially sublimation, are more interesting. Specifically, CRYOWRF produces higher total sublimation ($+129 \, \mathrm{Gt\,yr^{-1}}$) when compared to RpNew. While we observe in RpNew that the surface sublimation is reduced in the presence of blowing snow sublimation, such a trend is not visible in CRYOWRF results. Furthermore, the difference in erosion due to snow being blown off Antarctica is significant.

## 5 Summary and conclusions

In this study, we updated the blowing snow model in the regional climate model RACMO, version 2.3p3 (Rp3), to better represent the blowing snow phenomenon, the major ablation term in the SMB of the Antarctic ice sheet. As observed in the limited available observations, the unaltered version of the model Rp3 failed to accurately predict the power-law variation of blowing snow mass flux with wind speed compared to observations. Furthermore, choices made in the unaltered version to reduce the computational expenses of the blowing snow model led to simplifications and assumptions which affected the model results. In the present work, we updated the empirical formulation of saltation flux used as the boundary condition for the blowing snow model. We increased the number and distribution of ice-particle radius classes to cover all the relevant blowing snow radii classes. We also improved the coupling of the blowing snow model with RACMO by providing velocity, temperature profiles and friction velocities from RACMO to the blowing snow model, which was previously being modelled as a logarithmic-law velocity and the friction velocity was based on the first model level velocities. In addition, we found that the blowing snow model was very sensitive to its time step and introduced sub-stepping for the blowing snow model, which significantly improved the results.





We ran the original blowing snow model (Rp3) and the updated code (RpNew) for Antarctica on a 27 km grid laterally forced by 3-hourly ERA5 data. We performed three experiments for 2000–2010: Rp3, RpNew, and NO-DRIFT. In the last experiment, RACMO was run without the blowing snow model. The results from the updated model were evaluated against in-situ observation from site D47, Adélie Land, Antarctica (Amory, 2020). Important surface quantities such as the near-surface wind, temperature, humidity and blowing snow fluxes were compared. We found that RpNew results compared well against the blowing snow observations, successfully predicting both blowing snow frequency and magnitude. Furthermore, RpNew also successfully predicts the power-law variation of the blowing snow transport fluxes with wind speed. Comparison of continental blowing snow frequency obtained from RpNew with CALIPSO satellite observations (Palm et al., 2018) shows that qualitatively, RpNew predicts the blowing snow frequency over Antarctica reasonably well.

The updated estimates of blowing snow sublimation from RpNew also agree well with the continent-wide estimates of blowing snow sublimation from satellite observations. Average blowing snow depth of $230\pm116$ m obtained from RpNew matches reasonably well with the satellite observations from Palm et al. (2017). Furthermore, Palm et al. (2017) from CALIPSO lidar observations report a maximum blowing snow sublimation of $250\pm125$ mm w.e. yr$^{-1}$ near the Antarctic coast around $140°$E longitude. We observe a maximum blowing snow sublimation of $335\pm30$ mm w.e. yr$^{-1}$ at the location: $66.9°$S, $130.4°$E. CALIPSO satellite observations indicate blowing snow sublimation could be as high as $393\pm196$ Gt yr$^{-1}$. We observe a blowing snow sublimation of $176\pm10$ Gt yr$^{-1}$ with RpNew which shows there is a significant difference between model results and satellite observations. Palm et al. (2018) attribute the high blowing snow sublimation estimates to the errors associated with MERRA-2 reanalysis data (Gelaro et al., 2017) used for calculating sublimation, particle radius error, and extinction errors and therefore, the satellite estimates involve a large error. However, without other continental-scale estimates of blowing snow sublimation, future studies must properly document the differences between different methods.

We observe an interesting self-limiting nature of total sublimation from RACMO model results. Specifically, while the RpNew leads to an increase in the blowing snow sublimation, we observed a corresponding decrease in the surface sublimation and a non-negligible increase in deposition, balancing the total sublimation in Antarctic winter. Based on RpNew results, we hypothesise that sublimation in Antarctica is a self-limiting mechanism where large blowing snow sublimation saturates the near-surface layers, limiting the potential for surface sublimation. RpNew results support our hypothesis. However, results from the CRYOWRF model (Gerber et al., 2023), which also models the blowing snow phenomenon, do not follow this trend. Therefore, future intercomparison studies with other models are necessary to test the hypothesis.

Compared to Rp3, with an increase in the blowing snow sublimation due to higher moisture in the air, the deposition of water vapour at the surface increases, leading to only a small change in the overall sublimation in winter. Though overall trends in the sublimation between Rp3 and the RpNew appear to be the same, we observe a slight change in the climatology of blowing snow in the interior of Antarctica. In Rp3, the blowing snow sublimation was largely limited to the escarpment regions of Antarctica, with nearly zero blowing snow sublimation in the interior. RpNew results show that the results from Rp3 differ in both frequency and magnitude; the difference seems to be higher in the interior compared to the coastal Antarctica.

In conclusion, the updates introduced to the regional climate model RACMO in this study significantly improve the representation of blowing snow physics in RACMO. Blowing snow and surface sublimation are the major mass loss terms in the



SMB of Antarctica, leading locally to a negative SMB, which results in the formation of blue ice areas. This study presents a step forward in modelling blowing snow in producing a physically sound and reliable estimate of the SMB of Antarctica.

*Data availability.* Monthly and yearly averaged data for NO-DRIFT, 2.3p3, and updated versions of RACMO are available for the years 2000-2010.

*Author contributions.* SG and WJB conceived this study, decided on the new model settings. SG performed the code development and performed the model simulations and led the writing of the manuscript.

*Competing interests.* The authors declare that they have no conflict of interest.

*Acknowledgements.* We would like to thank Charles Amory for the discussion about observational dataset from site D47, East Antarctica. We would also like to thank Melchior van Wessem and Christiaan van Dalum for discussions about RACMO model development. This project has received funding from the European Union's Horizon 2020 research and innovation program under grant agreement no. 101003590 (PolarRES). We also acknowledge the ECMWF for archiving facilities and computational time on their supercomputers.





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
