# Peer review of "Contribution of blowing snow sublimation to the surface mass balance of Antarctica"

_EGUsphere, 2024_

## Referee Comment (RC1)

Review of "Contribution of blowing snow sublimation to the surface mass balance of Antarctica" by Gadde and van de Berg

**General comments**

The manuscript entitled "Contribution of blowing snow sublimation to the surface mass balance of Antarctica" by Gadde and van de Berg presents an update of the blowing snow model implemented in the regional climate model RACMO. The authors modified several equations and parametrisations. New model runs are compared to observational data from site D47 in Adélie Land, East Antarctica, to validate the results. The study highlights the importance of blowing snow sublimation to the surface mass balance (SMB) of the Antarctic Ice Sheet and provides a valuable contribution to better account for blowing snow sublimation in models. The addressed topic is within the scope of TC and discusses a relevant and current glaciological question.

Below, I provide some specific comments and suggestions for further improving the manuscript that should be addressed prior to publication in The Cryosphere.

**Specific comments (major)**

Overall, the manuscript is well structured and presents the changes made to the model as well as the results. The introduction ends with an overview of the content of the individual chapters and offers the reader a clear structure. However, the individual descriptions of the changes in the model and the results are very detailed and sometimes lengthy. In general, the text could be shortened and formulated more precisely in many places. Repetitions occur in various sections but should be avoided. The discussion is too brief, and the results of this specific model are only briefly compared with another model (section 4.6). This would be an interesting comparison and further validation of the results presented here. Unfortunately, the manuscript falls short on this comparison and the main conclusions, while other descriptions are very detailed. I suggest shortening the manuscript (especially chapter 4) and discussing the relevance and implications of the study in more detail. Furthermore, the language could benefit from proofreading.

The authors mention that the model runs are available, but it is not clear where the data can be found. I encourage the authors to make the data as well as the updated model code available via an open repository.

**Specific comments (minor)**

- L. 35: *continent-wise* – should it be *continent-wide*
- L. 40: RCM is already defined in l. 36. Please use abbreviations once they are introduced.

- L. 50: Are you referring to specific observations here or just generally saying that RACMO was evaluated against observations?
- L. 52: What is the difference between RACMO2.3p1 and p2 and why do the different versions suggest different blowing snow fluxes?
- L. 54: To which RACMO version are you referring here, i.e. which blowing snow module in RACMO?
- L. 64: which RACMO version? 2.3p2?
- L. 103: It should read: ..serves *as* the boundary condition.
- L. 111: It seems that the verbs are missing in this sentence.
- L. 113: Please rewrite this sentence; it is hard to follow.
- L. 123: It is again a long and nested sentence. It might be easier to follow shorter sentences.
- L. 125: How do you justify setting d = s = 0.5?
- L. 145: What is different in the PIEKTUK-D compared to the PIEKTUK model? Please explain.
- L. 147: It should read: ..follows *a* two-parameter gamma distribution.
- L. 164: I am missing a reasoning why you made exactly these six updates to the model. Can you provide a short explanation for that?
- L. 169: Please change the order of the Equations → (5) and (6)
- L. 170: Please mention and/or explain the entire method, not only mention the abbreviation DNS.
- L. 183: There is a *t* missing in constitutes.
- L. 192: What did you test in the sensitivity analysis? Did you compare the results to observations? How did you quantify that a time step of *10 seconds produces reliable estimates*?
- L. 214: Please provide references when mentioning, that it's widely used in the literature.
- L. 221: You could add a wind rose or another type of graphic to illustrate the directional consistency of the katabatic winds.
- L. 231: Please add a space character between *up* and *to*.
- L. 236: The description in this paragraph is a bit confusing for me. To clarify: The observations of wind speed are measured at a height of 2 m and the model results are obtained using the Monin-Obukhov theory to calculate from the first atmospheric level in RACMO (which height is this?) to a 2 m wind speed. Is this correct?
- L. 239: Which RACMO version was used for RpNew? RACMO2.3?
- L. 250: Are you referring to high annual mean wind speeds or high wind speeds during events? You introduced the data already in section 3. Please avoid describing the observational data at several places in the manuscript.
- L. 251: Are the values in the table mean values for the period 2010-2012? Is there a seasonality in the model-data agreement/disagreement?
- L. 252: I am not aware of the word *underprediction*. I would suggest using the word *underestimates* instead of *underpredict* here as well as in the rest of the manuscript. Same for *overpredicted* in l. 265.
- L. 256: Please provide a better description when you are talking about which model/model result.

- Figure 3: You are providing many numbers after the decimal point. I personally would suggest to only show two numbers to keep the plot simple and clear.
- L. 275: I agree that the modified model RpNew shows better agreement between observed and simulated values. However, if you write about significant improvements, I would like to see p-values and/or a measure of the significance of the results.
- L. 286: $R^2$ would be 0.57 if rounding from 0.5683 as given in Fig. 3e.
- L. 286: Please provide statistical evidence when mentioning significant improvements. Just mentioning an $R^2$ of 0.57 is not sufficient.
- L. 293: Here, you are referring to RACMO2.3p2. In the previous paragraphs, you often only mention RACMO. Are you referring to RACMO2.3p2 when writing RACMO? Please either specify each time the version you are referring to or mention once that you are referring to a specific version.
- L. 298-304: Please provide correlations, $R^2$ or another statistical measure to prove that RpNew predicts *reasonably well* the magnitude and occurrence of the blowing snow or that RpNew *successfully predicts blowing snow mass flux reliably*.
- L. 314: Please stay consistent throughout the manuscript and use Eq. instead of equation.
- L. 322: Again, please provide statistical evidence.
- L. 348-351: Are you referring to Antarctic winter or Nov – Jan winter months? Same for summer.
- L. 356: What is the reasoning that you investigate the year 2011 and not 2010 or 2012?
- L. 362: Please provide a height estimate for *the upper part of the boundary layer*.
- L. 368ff: The given study investigates an area with high katabatic winds and it is highlighted how important the wind speed estimation is. Please elaborate why this case study provides sufficient and reliable estimates to transfer and extent the results from this study to continent-wide estimations of blowing snow sublimation, especially in areas where the wind speeds are low throughout most of the year.
- Figure 6: Please provide a version of this plot with colourblind-friendly colours (for instance in the appendix).
- L. 386: You are mentioning that there are differences between the studies, but they are still comparable. Please, if you mention differences, then elaborate on them and provide a reasoning, why they are still qualitatively comparable.
- Figures 7 and 8: The colour scales are not colourblind friendly and, in several cases (e.g. Fig. 8e), it is hard to see which colour indicates 0.
- L. 434: How do you explain that your model has the maximum blowing snow sublimation slightly shifted in its location compared to the result from Palm et al.?
- L. 443: Remove *to* between *RpNew* and *lead*.
- L. 463: Please specify what months are considered as winter and summer.
- L. 477: Please quantify *large differences*.
- L. 495: Why is the period here 2000-2010 while the simulations are from 2000-2012?

- L. 511: Here, you mention that the results are for the period from 2000 to 2010. In l. 249, you are referring to a period from 2010 to 2012 which is confusing to me. Why are you using different time periods for different comparisons?
- L. 534: I am again confused by the time frame of the experiments. Here, the period 2000-2010 is mentioned while in l. 249 values are reported for the period 2010-2012. Please clarify.

---

## Author Comment (AC1)

**Response to RC1**

**Reviewer:** The manuscript entitled "Contribution of blowing snow sublimation to the surface mass balance of Antarctica" by Gadde and van de Berg presents an update of the blowing snow model implemented in the regional climate model RACMO. The authors modified several equations and parametrisations. New model runs are compared to observational data from site D47 in Adélie Land, East Antarctica, to validate the results. The study highlights the importance of blowing snow sublimation to the surface mass balance (SMB) of the Antarctic Ice Sheet and provides a valuable contribution to better account for blowing snow sublimation in models. The addressed topic is within the scope of TC and discusses a relevant and current glaciological question. Below, I provide some specific comments and suggestions for further improving the manuscript that should be addressed prior to publication in The Cryosphere.

We thank the reviewer for spending time on reviewing our manuscript. Below, we indicate the changes we intend to make in the manuscript.

**Reviewer:** Specific comments (major)

Overall, the manuscript is well structured and presents the changes made to the model as well as the results. The introduction ends with an overview of the content of the individual chapters and offers the reader a clear structure. However, the individual descriptions of the changes in the model and the results are very detailed and sometimes lengthy. In general, the text could be shortened and formulated more precisely in many places. Repetitions occur in various sections but should be avoided. The discussion is too brief, and the results of this specific model are only briefly compared with another model (section 4.6). This would be an interesting comparison and further validation of the results presented here. Unfortunately, the manuscript falls short on this comparison and the main conclusions, while other descriptions are very detailed. I suggest shortening the manuscript (especially chapter 4) and discussing the relevance and implications of the study in more detail. Furthermore, the language could benefit from proofreading. The authors mention that the model runs are available, but it is not clear where the data can be found. I encourage the authors to make the data as well as the updated model code available via an open repository.

**Response:** We thank the reviewer for the suggestions. We intend to shorten the manuscript in few sections elaborated below. To further compare CRYOWRF and RACMO, we will modify sections 4.4 and 4.5. In section 4.4, we will compare the sublimation patterns from CRYOWRF and RACMO. In section 4.5, we will add a comparison between the monthly blowing snow sublimation from CRYOWRF and RACMO, to further explore how blowing snow sublimation and surface sublimation vary within the two models.

Major changes envisioned include,

- We will shorten/rewrite sections 2.2 which includes the description of the blowing snow model.

- We will shorten section 4 by mergin sections 4.1.1 and 4.1.2. We will be refocusing the discussion in section 4.1.1 mostly on the comparison of relative humidity and blowing snow flux, since relative humidity is the most important quantity for sublimation and changes are observed in this. We will also shorten the text in these two sections.

- We will merge tables 2 and 3 in section 4.1.3.

- We will refocus the discussion on comparison between RpNew and NO-DRIFT instead of RpNew and Rp3. As such we intend to include the comparison betwen RpNew and Rp3 (section 4.3), in appendix.

- We will compare the sublimation pattern over Antarctica from CRYOWRF and RACMO in section 4.4.

- We will modify section 4.5 to include a comparison between monthly sublimation changes of CRY-OWRF and RACMO.

- We will also add an additional table in section 4.6 comparing RpNew and NO-DRIFT case to document the overall effect of blowing snow sublimation when compared to NO-DRIFT case.

Specific comments (minor)
- L. 35: continent-wise – should it be continent-wide
We will fix this during revision.

- L. 40: RCM is already defined in l. 36. Please use abbreviations once they are introduced.
We will modify this in the revised manuscript.

- L. 50: Are you referring to specific observations here or just generally saying that RACMO was evaluated against observations?
L50 is mentioning the evaluation in 'general'. Specifics are given in the next line. We will rephrase this to avoid confusion.

- L. 52: What is the difference between RACMO2.3p1 and p2 and why do the different versions suggest different blowing snow fluxes?
The updates included in RACMO2.3p2 are described in detail by van Wessem et al (The Cryosphere, 2018, https://doi.org/10.5194/tc-12-1479-2018). In brief, 2.3p2 was a retuning of RACMO in order to improve the spatial representation of the climate and surface mass balance for Greenland and Antarctica. Concerning snowdrift, in 2.3p2, the snowdrift flux was halved in order to bring the modelled snow drift fluxes more in line with the snow drift observations made in Greenland.

- L. 54: To which RACMO version are you referring here, i.e. which blowing snow module in RACMO?
We are referring to RACMO2.3p2, in which the saltation coefficient was halved, $c_{salt} = 0.192$.

- L. 64: which RACMO version? 2.3p2?
RACMO2.3p2.

- L. 103: It should read: ..serves as the boundary condition.
We will fix this in the revised manuscript.

- L. 111: It seems that the verbs are missing in this sentence.
We will rephrase this during revision.

- L. 113: Please rewrite this sentence; it is hard to follow.
We will rewrite this sentence.

- L. 123: It is again a long and nested sentence. It might be easier to follow shorter sentences.
We will rewrite this sentence.

- L. 125: How do you justify setting d = s = 0.5?
The variables $d$ and $s$ have been set following Lenaerts (2012) [1]. Our primary goal with this update was to remove the numerical artefact which resulted in inexplicable variation of blowing snow flux with wind speed. We found no indication that the mobility index needed to be retuned for removing the numerical artefact, hence the values of $d$ and $s$ were retained, as additional modifications catering to snow particle characteristics are out of the scope of the present study for reasons explained below.

The threshold friction velocity based on [2], depend on the snow mobility index which denotes the

potential for snow erosion by the wind, with $Mo = 0.75d - 0.5s + 0.5$, where d and s represent dendricity and sphericity of fresh snow. Galle et al. [2] mention that the crystal shape of fresh fallen snow does not allow a large grain cohesion in the snow pack. Therefore, this allows relatively high values of snow mobility index Mo for large $d$. Sintering is enhanced when the number of rounded shapes increases, so that $Mo$ decreases when $s$ decreases. Explicitly modelling the snow mobility index requires solving prognostic equations for snow particle characteristics, evaluation of which would require observational data. However, blowing snow frequency is more widely observed and the snow mobility index was set to 0.625 ($d = 0.5$ and $s = 0.5$) to match observations [1]. A detailed discussion of the same is available in Lenaerts et al. (2012) [1].

- L. 145: What is different in the PIEKTUK-D compared to the PIEKTUK model?
PIEKTUK model [3] is a single-moment model in which only a governing equation for blowing snow mixing ratio $q_b$ is solved. PIEKTUK-D is the 'double' moment model which has two governing equations for blowing snow quantities, one for $q_b$ and an additional equation for particle concentration $N$.

- L. 147: It should read: ..follows a two-parameter gamma distribution.
We will fix this in the revised manuscript.

- L. 164: I am missing a reasoning why you made exactly these six updates to the model. Can you provide a short explanation for that?
While analysing whether the updated PIEKTUK-D implementation in RACMO matches the results of the original offline PIEKTUK-D code, we found these exact issues. The six updates were needed to improve representation of snowdrift and its impact on the boundary layer in RACMO. As mentioned in the manuscript, in Rp3 many assumptions were made to simplify the implementation. These assumptions, though simplified the coupling PIEKTUK-D with RACMO, were not accurate. We made these changes to improve the coupling between RACMO and the blowing snow module and to make the calculations of PIEKTUK-D correct.

- L. 169: Please change the order of the Equations à (5) and (6)
We will fix this in the revised manuscript.

- L. 170: Please mention and/or explain the entire method, not only mention the abbreviation DNS.
Thanks for noticing. It is 'direct numerical simulations'. We will fix this in the revised manuscript.

- L. 183: There is a t missing in constitutes.
We will fix this in the revised manuscript.

- L. 192: What did you test in the sensitivity analysis? Did you compare the results to observations? How did you quantify that a time step of 10 seconds produces reliable estimates?
We carried out 'time step' sensitivity test. Blowing snow model reaches equilibrium quickly after initialisation, however the equilibrium value of the blowing snow mixing ratio and particle concentration depends on the time step used for solving the evolution equations 5 and 6 (for $q_b$ and $N$). We ran the model with different $\Delta t$, specifically, with default RACMO time step 600 s and reduced time step sizes. The magnitude of vertically integrated blowing snow flux was checked for each simulation. We found large difference between the values for $\Delta t = 600, 300, 100, 50$ and $20$. For $\Delta t = 10$ and $\Delta t = 5$, the magnitude of blowing snow flux was nearly the same, however $\Delta t = 5$ required more number of sub-steps to reach the value. So we chose $\Delta t = 10$ s.
We will briefly mention this in the revised manuscript.

- L. 214: Please provide references when mentioning, that it's widely used in the literature.

We will add the references in the revised manuscript.

- L. 221: You could add a wind rose or another type of graphic to illustrate the directional consistency of the katabatic winds.
We did not add the wind rose as the same is already available with the observational dataset [4, 5]. Specifically, Figure S1 in supplement [4]. We will add reference to this.

- L. 231: Please add a space character between up and to.
We will fix this in the revised manuscript.

- L. 236: The description in this paragraph is a bit confusing for me. To clarify: The observations of wind speed are measured at a height of 2 m and the model results are obtained using the Monin-Obukhov theory to calculate from the first atmospheric level in RACMO (which height is this?) to a 2 m wind speed. Is this correct?
Yes, that is correct. We will rephrase it to make it clear.

- L. 239: Which RACMO version was used for RpNew? RACMO2.3?
RACMO2.3p3.

- L. 250: Are you referring to high annual mean wind speeds or high wind speeds during events? You introduced the data already in section 3. Please avoid describing the observational data at several places in the manuscript.
We meant high annual mean wind speeds here. We will rewrite this to avoid repetition.

- L. 251: Are the values in the table mean values for the period 2010-2012? Is there a seasonality in the model-data agreement/disagreement?
The values in the table represent the values of linear regression between observations and model results. We will explore the seasonality and include it in the manuscript.

- L. 252: I am not aware of the word underprediction. I would suggest using the word underestimates instead of underpredict here as well as in the rest of the manuscript. Same for overpredicted in l. 265.
We will rephrase this in the revised manuscript to avoid confusion.

- L. 256: Please provide a better description when you are talking about which model/model result.
Since both Rp3, RpNew, and NO-DRIFT predict similar values for wind speed we just mentioned RACMO. We will rephrase this in the revised manuscript.

- Figure 3: You are providing many numbers after the decimal point. I personally would suggest to only show two numbers to keep the plot simple and clear.
We will modify this in the revised manuscript.

- L. 275: I agree that the modified model RpNew shows better agreement between observed and simulated values. However, if you write about significant improvements, I would like to see p-values and/or a measure of the significance of the results.
For all the linear regressions presented in the manuscript, p-value was used to determine if the correlations are significant. We will modify the manuscript to specify the p-value of regressions. We will also rephrase/adapt the manuscript when necessary. For all the regressions we obtained a $p-value < 0.01$.

- L. 286: R2 would be 0.57 if rounding from 0.5683 as given in Fig. 3e.
We will fix this in the revised manuscript.

- L. 286: Please provide statistical evidence when mentioning significant improvements. Just mentioning an R2 of 0.57 is not sufficient.

For all the linear regressions presented in the manuscript, p-value was used to determine if the correlations are significant. We will modify the manuscript to specify the p-value of regressions. We will also rephrase/adapt the manuscript when necessary. For all the regressions we obtained a $p-value < 0.01$.

- L. 293: Here, you are referring to RACMO2.3p2. In the previous paragraphs, you often only mention RACMO. Are you referring to RACMO2.3p2 when writing RACMO? Please either specify each time the version you are referring to or mention once that you are referring to a specific version.

Sorry about the confusion, we will carefully rephrase this in the revised manuscript.

- L. 298-304: Please provide correlations, R2 or another statistical measure to prove that RpNew predicts reasonably well the magnitude and occurrence of the blowing snow or that RpNew successfully predicts blowing snow mass flux reliably.

For all the linear regressions presented in the manuscript, p-value was used to determine if the correlations are significant. We will modify the manuscript to specify the p-value of regressions. We will also rephrase/adapt the manuscript when necessary. For all the regressions we obtained a $p-value < 0.01$.

- L. 314: Please stay consistent throughout the manuscript and use Eq. instead of equation.

We will fix this in the revised manuscript.

- L. 322: Again, please provide statistical evidence.

For all the linear regressions presented in the manuscript, p-value was used to determine if the correlations are significant. We will modify the manuscript to specify the p-value of regressions. We will also rephrase/adapt the manuscript when necessary. For all the regressions we obtained a $p - value < 0.01$..

- L. 348-351: Are you referring to Antarctic winter or Nov – Jan winter months? Same for summer.

We refer to Antarctic winter, we will specify this in the revised manuscript.

- L. 356: What is the reasoning that you investigate the year 2011 and not 2010 or 2012?

The 'original' height of the sensor at D47 is 2.8 m. However, Amory et al. (2020) [4] mention that, due to harsh weather conditions at D47, they could not do reset the height of the sensors owing to the elevation changes due to snow. As a result, by late December 2012, the measurement heights decreased from their initial values to 1.5 m for wind speed and direction and 0.9 m for temperature and relative humidity. We decided to do the evaluation for 2011, since the 'sensor' height in this year was around 2 m, and was easier to compare with the model results.

We will add this information in the revised manuscript.

- L. 362: Please provide a height estimate for the upper part of the boundary layer.

We will include the planetary boundary layer height in the revised manuscript.

- L. 368Q: The given study investigates an area with high katabatic winds and it is highlighted how important the wind speed estimation is. Please elaborate why this case study provides sufficient and reliable estimates to transfer and extent the results from this study to continent-wide estimations of blowing snow sublimation, especially in areas where the wind speeds are low throughout most of the year.

Please note that our primary goal was to correct the variation of snowdrift flux as function of the near surface wind speed, the goal was not to get the best agreement with the snow drift observations on the expense of the relation between wind speed and snow drift flux. Since, the blowing snow flux now 'varies' in the expected, 'power-law' fashion we believe it provides a better overall representation of the fluxes.

It is worth mentioning here that, we are not aware of any other dataset of snow drift observations in Antarctica of such high quality, therefore this is the best comparison that we could do. Furthermore, as snow drift fluxes are increasing exponentially with the wind speed, big errors are potentially made in high-windspeed regions. The impact of snowdrift sublimation in regions with low wind speeds is inherently small, even though these regions are much larger in size.

Figure 6: Please provide a version of this plot with colourblind-friendly colours (for instance in the appendix).

We will include a colourblind-friendly version of the figure in the revised manuscript.

L. 386: You are mentioning that there are differences between the studies, but they are still comparable. Please, if you mention differences, then elaborate on them and provide a reasoning, why they are still qualitatively comparable.

By differences we meant the different simulation time period 2010-2020 (CRYOWRF), while 2000-2012 for RACMO. Longer simulations with RACMO2.3p2 (van Wessem et al) and RACMO2.4p1 (van Dalum et al 2024) show that the Antarctic climate of 2013-2020 is very comparable to it of the period 2000-2012. Therefore, we mentioned that they are still comparable. We will rephrase this to be more precise.

- Figures 7 and 8: The colour scales are not colourblind friendly and, in several cases (e.g. Fig. 8e), it is hard to see which colour indicates 0.

We will replot the figures with colourblind friendly colormaps.

- L. 434: How do you explain that your model has the maximum blowing snow sublimation slightly shifted in its location compared to the result from Palm et al.?

Palm et al. (2018) results are mostly available for cloud-free skies and blowing snow layer height greater than 30 m. Since our main goal was to compare the results with the observations from site D47, we initially did not think about filtering blowing snow fluxes to only 'cloud-free' days and blowing snow layers of height greater than 30 m. Since we did not store the multi-level data required for filtering results, it would be difficult to ascertain the reason for the slight shift. Therefore, we decided to only look at the seasonal patterns and 'qualitatively' compare our results, since it was not our goal to perform a one-to-one comparison of our results with the satellite observations. We will mention the limitation in the revised manuscript in detail.

- L. 443: Remove to between RpNew and lead.

We will fix this in the revised manuscript.

- L. 463: Please specify what months are considered as winter and summer.

We will rewrite this in the revised manuscript.

- L. 477: Please quantify large differences.

It is mentioned in detail in Table 4.6, there is a 41% increase in the blowing snow sublimation with RpNew when compared to Rp3. We will rephrase the sentence to be more precise.

- L. 495: Why is the period here 2000-2010 while the simulations are from 2000- 2012?

Main reason is to present the results of the average over a decade, so we decided to leave 2011-2012 from the Yearly averages.

- L. 511: Here, you mention that the results are for the period from 2000 to 2010. In l. 249, you are referring to a period from 2010 to 2012 which is confusing to me. Why are you using different time periods for dieffernt comparisons?

See previous response. We will also present the yearly average for 2010-2012 to avoid confusion.

- L. 534: I am again confused by the time frame of the experiments. Here, the period 2000-2010 is mentioned while in l. 249 values are reported for the period 2010-2012. Please clarify.
See previous response. We will also present the yearly average for 2010-2012 to avoid confusion.

**References**

[1] J. T. M. Lenaerts, M. R. van den Broeke, S. J. Déry, E. van Meijgaard, W. J. van de Berg, S. P. Palm, and J. Sanz Rodrigo. Modeling drifting snow in antarctica with a regional climate model: 1. methods and model evaluation. *Journal of Geophysical Research: Atmospheres*, 117(D5), 2012.

[2] H. Gallée, G. Guyomarćh, and E. Brun. Impact of snow drift on the antarctic ice sheet surface mass balance: possible sensitivity to snow-surface properties. *Boundary-Layer Meteorology*, 99:1–19, 2001.

[3] S. J. Déry and M. K. Yau. A bulk blowing snow model. *Boundary-Layer Meteorology*, 93:237–251, 1999.

[4] C. Amory. Drifting-snow statistics from multiple-year autonomous measurements in adélie land, east antarctica. *The Cryosphere*, 14(5):1713–1725, 2020.

[5] C. Amory, C. Genthon, and V. Favier. A drifting snow data set (2010-2018) from coastal adelie land, eastern antarctica. *Data*, 2020.

---

## Author Comment (AC2)

**Response to RC2**

**Reviewer:** The authors introduce a new version of the blowing snow scheme in RACMO. The new model is described and a first validation comparing the model to local as well as satellite blowing snow measurements is provided. Furthermore, the authors also discuss their results in comparison to an older version of RACMO and another continent-wide simulation over Antarctica. The paper overall fits the scope of the journal. Below a list of comments if provided, which should be taken into account before considering the paper for publication.
**Response:** We thank the reviewer for spending time on reviewing our manuscript. Below, we indicate the changes we intend to make in the manuscript.

General / major comments:
For the validation against measurements as well as compared to the other model, the term "significant" is used regularly in the text. It is unclear if significance tests have been performed. Are the changes actually significant based on significance tests? And if yes, which changes are and which are not? If this is not feasible at least the wording should be adapted accordingly (see detailed comments below).
For all the linear regressions presented in the manuscript, p-value was used to determine if the correlations are significant. We will modify the manuscript to specify the p-value of regressions. We will also rephrase/adapt the manuscript when necessary.

The authors state, that there are differences between the satellite based estimate of blowing snow frequency and the model based results due to the fact that on one hand the satellite based results only start at 30 m above ground, while the results for RACMO include all the layers close to the surface. Furthermore, CALIPSO can only see blowing snow during cloud-free conditions or for optically thin clouds. Did the authors consider excluding cloudy days and the model levels below 30 m for the analysis? What is the reason for not applying these corrections to the displayed result? This would give the comparison more weight, as then the results could more directly be compared (relates to P16, L380ff).
Main goal of our study was to fix the numerical artefact in the variation of blowing snow fluxes with wind speed and subsequently to evaluate it by comparing the blowing snow results with the observations from D47, Antarctica. Therefore, we did not store the multi-level blowing snow transport data (or filtered data above 30 m). We decided to do a 'qualitative' comparison with Palm et al. (2018) after the runs were done. Therefore, we had to only look at the seasonal patterns and 'qualitatively' compare our results, since it was not our goal to perform a one-to-one comparison of our results with the satellite observations. Since Palm et al. (2018) [1] observations are limited to mostly optically thin cloud conditions, we believe the observations do not give the complete picture of the blowing snow events, and we intended to report the blowing snow events as we see from the model results. Simply put, with the current data output, it is impossible to perform the said filtering. Nonetheless, we tried to filter the blowing snow flux with the cloud fraction, but as expected this yields high blowing snow frequency as it is the total integrated blowing snow flux from all the layers.
We will clarify this in the revised manuscript.

Minor comments:
P1, L7: In "a non-uniformly discretized ice particle radii to . . . " radii - radius
We will fix this in the revised manuscript.

P4, L85: TrasnfEr - Transfer
We will fix the typo during revision.

P5, L113: empirircal - empirical
We will fix this during revision.

[Figure]

Figure 1: Threshold friction velocity for Rp3 and RpNew. Time is shown in month.

P5, L114: ", is" does not seem to fit the sentence. Remove ", is" or reformulate.
We will reformulate this during revision.

P9, L223: Is wind speed measured at 2m above ground? Based on Amory, 2020 it seems to be at 2.8 m originally for station D-47. Is 2m an average over the measurement period given the elevation changes due to snow? (if a change is required take it into account accordingly for P10, L261).
Your observation is correct that the 'original' height of the sensor at D47 is 2.8 m. However, in the paper they mention that, due to harsh weather conditions at D47, they could not do reset the height of the sensors. As a result, by late December 2012, the measurement heights decreased from their initial values to 1.5 m for wind speed and direction and 0.9 m for temperature and relative humidity. The period we considered for evaluation i.e. 2011, the height was reduced from its initial 2.8 m and was lower. And the processed data that is made available on Zenodo [2], the height of the measurement is mentioned as 2 m. Furthermore, in the supplementary materials of the data paper [3] they mention that mean values for wind speed, temperature, and relative humidity are determined from the measurement level closest to 2 m.
We will also mention this in the revised manuscript.

P11: In the comparison between the modelled and the measured data, it is shown that the results improve when simulating with blowing snow and even more when using the new version. Especially in lines 270ff the authors talk about significant biases and improvements. Are these statements based on a significance test? What kind of test was used? – If not either the wording should be changed or a significance test should be performed to confirm. In general, I think it would be interesting to see which improvements are significant.
For all the linear regressions presented in the manuscript, all the regression coefficients are tested for statistical signficiance using p-value. Specifically, for all the regressions p-value < 0.01. We will modify the manuscript to specify the p-value of regressions. We will also rephrase/adapt the manuscript if necessary.

P14, Section 4.1.3: It is discussed that Rp3 overall shows a better performance in predicting the blowing snow frequency overall, while RpNew mainly improves the prediction of higher magnitude blowing snow events. It is great to see that RpNew improves for the higher magnitude events. It would, however, be interesting to see both effects discussed? Why is there an improvement for the higher magnitude events and why does it perform worse overall? What are the impacts of the decrease in performance when looking at the overall blowing snow frequency.
We will add additional discussion about why RpNew predicts lower number of events.

This is due to the comparatively 'better' performance of Rp3 during the Antarctic summer. Specifically, in Rp3, instead of using the friction velocity $u_*$, calculated in the land surface scheme, $u_*$ was recalculated in every step with a simple log-law assumption (with neutral stratification). Since the assumption was not correct, we changed it and used the friction velocity calculated from the physics module. Simply put, the assumption which was previous used (though not correct) reduced the threshold friction velocity, and thereby we observed more blowing snow events with Rp3. Figure 1 shows the threshold friction velocity used in Rp3 and RpNew, it is clear from the figure that assumptions and simplifications made in Rp3 reduced the threshold friction velocity overall which would result in higher blowing snow events compared to RpNew.

Towards overall blowing snow frequency, preliminary analysis tells us that at site D47, this behaviour is mostly seasonal, in Antarctic winter when the wind speeds are high,the blowing snow events are predicted well (as you can see in Fig. 4 of the manuscript). The decreased performance is mostly in summer, however this needs to be quantified.

We will calculate the blowing snow events in Antarctic winter and summer for both Rp3 and RpNew and include this information in the revised manuscript.

P15, L367: "... 5(a)).This analysis..." - Missing space after.
We will fix this in the revised manuscript.

P21, L476: "...in under-prediction blowing snow particle concentration." - Should probably read: "...in an under-prediction of the blowing snow particle concentration."
We will fix this in the revised manuscript.

P22, Table 4: From just reading the table caption and the formula, it is not clear that ERds only refers to the eroded snow blown off the continent. Please clarify.
ERds only considers the transport aspect of snowdrift. ERds is positive in case of erosion due to divergence of the snowdrift flux, and negative if convergence of the snowdrift flux brings snow to a grid box. So when ERds is negative at but less than the snow drift sublimation, snow drift still has a net erosion effect.
Furthermore, as ERds only considers the snow redistribution, the spatially integrated impact on the SMB is zero as long as drifting snow is not blown of the ice sheet. That is the reason why ERds is locally very important, but integrated over the ice sheet insignificant.
We will clarify this in the revised manuscript.

P22, Table 4: For "(a) Difference between RpNew and RP3, and (b) SMB difference between RpNew (2000-2010) with CRYOWRF (2010–2020)": Add information about the two differences shown. E.g. ... in Gt (% of xxx). To which value is the percentage related?
It is the percentage change in the quantities compared to previous version i.e. Rp3. It is calculated as the (Rp3 - RpNew)/(Rp3). The word 'difference' might not be accurate here, we will rewrite this during revision.

P22, Table4b and P23, L509ff: The time periods of the RACMO and CRYOWRF simulations are not the same (RACMO: 2000-2010; CRYOWRF: 2010-2020). The difference in the between the simulated time periods is not discussed. How much might the different periods influence the difference in the results between the two models?
Our experience with RACMO runs suggest that the total sublimation does not vary much in the decade between 2000-2020. Therefore, we do not expect a large difference in the results, especially sublimation, during this time period.
To elaborate, after the submission of the current manuscript, the blowing snow updates along with other Physics updates were introduced in RACMO, version 2.4p1 [4]. Figure 2 shows the yearly average of

[Figure]

Figure 2: Variation of SMB and its components from RACMO2.4, newest version of RACMO with additional physics update. The figure is a part of the manuscript in preparation.

SMB components for RACMO 2.4p1. The figure is a part of the manuscript in preparation. As can be seen from the figure the sublimation (SUSD) predicted by RACMO2.4p1 does not vary much between 2000-2020, and therefore we think it is justified to compare the results.
We will mention the difference in the time-periods and its possible influence in the revised manuscript.

P23, L516ff: The authors state that they can see a trend that the surface sublimation is reduced in the presence of blowing snow sublimation. This is very interesting, can this hypothesis be discussed in more detail. And as a second step, how can it be shown that this trend is not present in CRYOWRF, compared to what? (Same for P24, L556ff).
This can be further explored by comparing the blowing snow sublimation for RpNew or Rp3 with NO-DRIFT case. We focused more on comparing RpNew and Rp3 in the initial version of the manuscript and therefore, we did not include the surface sublimation plots for the NO-DRIFT case in Figure 9. In the revised manuscript we will add additional discussion related to this.
On observing this trend with CRYOWRF, it would be difficult to show this without a simulation of CRYOWRF with the blowing snow model switched-off. However, the data provided by Gerber et al. (2023) does not include a simulation without the blowing snow sublimation. However, we will include monthly sublimation from CRYOWRF in our plots (see Figure 3), from the figure we see that major difference between Rp3 and RpNew comes in winter months when blowing snow sublimation is dominant. Increase in blowing snow sublimation of around 6 Gt/yr in winter is balanced by the 'condensation' of approximately 6 Gt/yr. As you can see in the figure, CRYOWRF does not show such a 'condensation' as the surface sublimation during winter from CRYOWRF is nearly zero. This indicates that such an effect is not present within CRYOWRF. But, for a definitive answer, we would need a simulation of CRYOWRF without the blowing snow model, and can be explored in the future.
We will add a discussion related to this in the revised manuscript.

P23, L518: The authors state that the difference in erosion due to snow being blown off Antarctica is "significant". Is this based on a significance tests? What test has been used? Please specify or reformulate.

[Figure]

Figure 3: Monthly variation of surface-, blowing snow, and total sublimation.

We will reformulate the sentence.

P23, L527: I think it should read "radius classes" instead of "radii classes"
We will fix this during revision.

P24, L536: "insitu observation from site D47": observation -¿ observations
We will correct this during revision.

P25, L570: The authors state that the RACMO data is available, but they don't say where. Please add a link to the repository from which the data can be downloaded.
We will upload the datasets on public repository before final revision and provide a DOI to the same.

**References**

[1] S. P. Palm, V. Kayetha, and Y. Yang. Toward a satellite-derived climatology of blowing snow over antarctica. *Journal of Geophysical Research: Atmospheres*, 123(18):10–301, 2018.
[2] C. Amory, C. Genthon, and V. Favier. A drifting snow data set (2010-2018) from coastal adelie land, eastern antarctica. *Data*, 2020.
[3] C. Amory. Drifting-snow statistics from multiple-year autonomous measurements in adélie land, east antarctica. *The Cryosphere*, 14(5):1713–1725, 2020.
[4] Christiaan T van Dalum, Willem Jan van de Berg, Srinidhi N Gadde, Maurice van Tiggelen, Tijmen van der Drift, Erik van Meijgaard, Lambertus H van Ulft, and Michiel R van den Broeke. First results of the polar regional climate model racmo2. 4. *EGUsphere*, 2024:1–36, 2024.

---

## Author Response (AR1)

**Response to RC1**

**Reviewer:** The manuscript entitled "Contribution of blowing snow sublimation to the surface mass balance of Antarctica" by Gadde and van de Berg presents an update of the blowing snow model implemented in the regional climate model RACMO. The authors modified several equations and parametrisations. New model runs are compared to observational data from site D47 in Adélie Land, East Antarctica, to validate the results. The study highlights the importance of blowing snow sublimation to the surface mass balance (SMB) of the Antarctic Ice Sheet and provides a valuable contribution to better account for blowing snow sublimation in models. The addressed topic is within the scope of TC and discusses a relevant and current glaciological question. Below, I provide some specific comments and suggestions for further improving the manuscript that should be addressed prior to publication in The Cryosphere.

We thank the reviewer for spending time on reviewing our manuscript. Below, we indicate the changes made to the manuscript.

**Reviewer:** Specific comments (major)

Overall, the manuscript is well structured and presents the changes made to the model as well as the results. The introduction ends with an overview of the content of the individual chapters and offers the reader a clear structure. However, the individual descriptions of the changes in the model and the results are very detailed and sometimes lengthy. In general, the text could be shortened and formulated more precisely in many places. Repetitions occur in various sections but should be avoided. The discussion is too brief, and the results of this specific model are only briefly compared with another model (section 4.6). This would be an interesting comparison and further validation of the results presented here. Unfortunately, the manuscript falls short on this comparison and the main conclusions, while other descriptions are very detailed. I suggest shortening the manuscript (especially chapter 4) and discussing the relevance and implications of the study in more detail. Furthermore, the language could benefit from proofreading. The authors mention that the model runs are available, but it is not clear where the data can be found. I encourage the authors to make the data as well as the updated model code available via an open repository.

**Response:** We thank the reviewer for the suggestions.

Major changes in the updated manuscript are,

- Section 2.2, including the description of the blowing snow model, is rewritten.

- Both sections 4.1.1 and 4.1.2 have been shortened. The discussion of section 4.1.1 has been shortened and refocused on the evaluation of blowing snow flux and relative humidity. The evaluation of the related near-surface quantities, such as wind speed, temperature, and specific humidity, are given in the appendix. Section 4.1.2 is also shortened.

- Tables 2 and 3 in section 4.1.3 are merged.

- The manuscript is now rewritten to discuss the effect of blowing snow compared to a NO-DRIFT case scenario. Results in section 4.3 focus now on RpNew and NO-DRIFT instead of RpNew and Rp3. And the comparison between RpNew and Rp3 (section 4.3), in the appendix.

- We have performed additional analysis and comparison of monthly variation of the sublimation between CRYOWRF and RACMO in section 4.5.

- We have added a table in section 4.6 comparing the RpNew and NO-DRIFT cases to document the overall effect of blowing snow sublimation compared to the NO-DRIFT case.

- L. 35: continent-wise – should it be continent-wide
Fixed in the revised manuscript.

- L. 40: RCM is already defined in l. 36. Please use abbreviations once they are introduced.
Fixed in the revised manuscript.

- L. 50: Are you referring to specific observations here or just generally saying that RACMO was evaluated against observations?
L50 is mentioning the evaluation in 'general'. Specifics are given in the next line. Rephrased in the revised manuscript.

- L. 52: What is the difference between RACMO2.3p1 and p2 and why do the different versions suggest different blowing snow fluxes?
The updates included in RACMO2.3p2 are described in detail by van Wessem et al (The Cryosphere, 2018, https://doi.org/10.5194/tc-12-1479-2018). In brief, 2.3p2 was a retuning of RACMO in order to improve the spatial representation of the climate and surface mass balance for Greenland and Antarctica. Concerning snowdrift, in 2.3p2, the snowdrift flux was halved in order to bring the modelled snow drift fluxes more in line with the snow drift observations made in Greenland.

- L. 54: To which RACMO version are you referring here, i.e. which blowing snow module in RACMO?
We are referring to RACMO2.3p2, in which the saltation coefficient was halved, $c_{salt} = 0.192$.

- L. 64: which RACMO version? 2.3p2?
RACMO2.3p2.

- L. 103: It should read: ..serves as the boundary condition.
Fixed in the revised manuscript.

- L. 111: It seems that the verbs are missing in this sentence.
Fixed in the revised manuscript..

- L. 113: Please rewrite this sentence; it is hard to follow.
Fixed in the revised manuscript.

- L. 123: It is again a long and nested sentence. It might be easier to follow shorter sentences.
Fixed in the revised manuscript.

- L. 125: How do you justify setting d = s = 0.5?
The variables $d$ and $s$ have been set following Lenaerts (2012) [1]. Our primary goal with this update was to remove the numerical artefact which resulted in inexplicable variation of blowing snow flux with wind speed. We found no indication that the mobility index needed to be retuned for removing the numerical artefact, hence the values of $d$ and $s$ were retained, as additional modifications catering to snow particle characteristics are out of the scope of the present study for reasons explained below.

The threshold friction velocity based on [2], depend on the snow mobility index which denotes the potential for snow erosion by the wind, with $Mo = 0.75d - 0.5s + 0.5$, where d and s represent dendricity and sphericity of fresh snow. Galle et al. [2] mention that the crystal shape of fresh fallen snow does not allow a large grain cohesion in the snow pack. Therefore, this allows relatively high values of snow mobility index Mo for large $d$. Sintering is enhanced when the number of rounded shapes increases, so that $Mo$ decreases when $s$ decreases. Explicitly modelling the snow mobility index requires solving prognostic equations for snow particle characteristics, evaluation of which would require observational data. However, blowing snow frequency is more widely observed and the snow mobility index was set to 0.625 ($d = 0.5$ and $s = 0.5$) to match observations [1]. A detailed discussion of the same is available in Lenaerts et al. (2012) [1].

- L. 145: What is different in the PIEKTUK-D compared to the PIEKTUK model?
PIEKTUK model [3] is a single-moment model in which only a governing equation for blowing snow mixing ratio $q_b$ is solved. PIEKTUK-D is the 'double' moment model which has two governing equations for blowing snow quantities, one for $q_b$ and an additional equation for particle concentration $N$.

- L. 147: It should read: ..follows a two-parameter gamma distribution.
Fixed in the revised manuscript.

- L. 164: I am missing a reasoning why you made exactly these six updates to the model. Can you provide a short explanation for that?
While analysing whether the updated PIEKTUK-D implementation in RACMO matches the results of the original offline PIEKTUK-D code, we found these exact issues. The six updates were needed to improve representation of snowdrift and its impact on the boundary layer in RACMO. As mentioned in the manuscript, in Rp3 many assumptions were made to simplify the implementation. These assumptions, though simplified the coupling PIEKTUK-D with RACMO, were not accurate. We made these changes to improve the coupling between RACMO and the blowing snow module and to make the calculations of PIEKTUK-D correct.

- L. 169: Please change the order of the Equations à (5) and (6)
Fixed in the revised manuscript.

- L. 170: Please mention and/or explain the entire method, not only mention the abbreviation DNS. It is 'direct numerical simulations'. Fixed in the revised manuscript.

- L. 183: There is a t missing in constitutes.
Fixed in the revised manuscript.

- L. 192: What did you test in the sensitivity analysis? Did you compare the results to observations? How did you quantify that a time step of 10 seconds produces reliable estimates?
We carried out 'time step' sensitivity test. Blowing snow model reaches equilibrium quickly after initialisation, however the equilibrium value of the blowing snow mixing ratio and particle concentration depends on the time step used for solving the evolution equations 5 and 6 (for $q_b$ and $N$). We ran the model with different $\Delta t$, specifically, with default RACMO time step 600 s and reduced time step sizes. The magnitude of vertically integrated blowing snow flux was checked for each simulation. We found large difference between the values for $\Delta t = 600, 300, 100, 50$ and 20. For $\Delta t = 10$ and $\Delta t = 5$, the magnitude of blowing snow flux was nearly the same, however $\Delta t = 5$ required more number of sub-steps to reach the value. So we chose $\Delta t = 10$ s.
We have added this description in Section 2.3.

- L. 214: Please provide references when mentioning, that it's widely used in the literature.
We have added a reference in the revised manuscript.

- L. 221: You could add a wind rose or another type of graphic to illustrate the directional consistency of the katabatic winds.
We did not add the wind rose as the same is already available with the observational dataset [4, 5].

Specifically, Figure S1 in supplement [4]. We will add reference to this.

- L. 231: Please add a space character between up and to.
Fixed in the revised manuscript.

- L. 236: The description in this paragraph is a bit confusing for me. To clarify: The observations of wind speed are measured at a height of 2 m and the model results are obtained using the Monin-Obukhov theory to calculate from the first atmospheric level in RACMO (which height is this?) to a 2 m wind speed. Is this correct?
Yes, that is correct. Rewritten in the revised manuscript.

- L. 239: Which RACMO version was used for RpNew? RACMO2.3?
RACMO2.3p3.

- L. 250: Are you referring to high annual mean wind speeds or high wind speeds during events? You introduced the data already in section 3. Please avoid describing the observational data at several places in the manuscript.
We meant high annual mean wind speeds here. We have rephrased this in the revised manuscript.

- L. 251: Are the values in the table mean values for the period 2010-2012? Is there a seasonality in the model-data agreement/disagreement?
The values in the table represent the linear regression values between observations and model results. We did not observe any seasonality in the model data; the model performance remains the same.

- L. 252: I am not aware of the word underprediction. I would suggest using the word underestimates instead of underpredict here as well as in the rest of the manuscript. Same for overpredicted in l. 265.
Rephrased in the revised manuscript.

- L. 256: Please provide a better description when you are talking about which model/model result.
Since both Rp3, RpNew, and NO-DRIFT predict similar values for wind speed we just mentioned RACMO. Fixed in the revised manuscript.

- Figure 3: You are providing many numbers after the decimal point. I personally would suggest to only show two numbers to keep the plot simple and clear.
Only two digits are shown after the deimal point. Fixed in the revised manuscript.

- L. 275: I agree that the modified model RpNew shows better agreement between observed and simulated values. However, if you write about significant improvements, I would like to see p-values and/or a measure of the significance of the results.
For all the linear regressions presented in the manuscript, p-value was used to determine if the correlations are significant. We will modify the manuscript to specify the p-value of regressions. We will also rephrase/adapt the manuscript when necessary. For all the regressions we obtained a $p-value < 0.01$.

- L. 286: R2 would be 0.57 if rounding from 0.5683 as given in Fig. 3e.
Fixed in the revise manuscript.

- L. 286: Please provide statistical evidence when mentioning significant improvements. Just mentioning an R2 of 0.57 is not sufficient.
For all the linear regressions presented in the manuscript, p-value was used to determine if the correlations are significant. We will modify the manuscript to specify the p-value of regressions. We will

also rephrase/adapt the manuscript when necessary. For all the regressions we obtained a $p-value < 0.01$.

In the manuscript, we have mentioned the particular version of RACMO wherever it was unclear.

For all the linear regressions presented in the manuscript, p-value was used to determine if the correlations are significant. We will modify the manuscript to specify the p-value of regressions. We will also rephrase/adapt the manuscript when necessary. For all the regressions we obtained a $p-value < 0.01$.

Fixed in the revised manuscript.

For all the linear regressions presented in the manuscript, p-value was used to determine if the correlations are significant. We will modify the manuscript to specify the p-value of regressions. We will also rephrase/adapt the manuscript when necessary. For all the regressions we obtained a $p-value < 0.01$..

We refer to Antarctic winter, we have now specified this in the manuscript.

The 'original' height of the sensor at D47 is 2.8 m. However, Amory et al. (2020) [4] mention that, due to harsh weather conditions at D47, they could not do reset the height of the sensors owing to the elevation changes due to snow. As a result, by late December 2012, the measurement heights decreased from their initial values to 1.5 m for wind speed and direction and 0.9 m for temperature and relative humidity. We decided to do the evaluation for 2011, since the 'sensor' height in this year was around 2 m, and was easier to compare with the model results.
We have added this information in the revised manuscript.

In L 362, we intended to mention that blowing snow sublimation does not happen at the surface as the air is saturated near the surface but happens above the surface in the boundary layer. We did not mean it is the top of the planetary boundary layer. We have rephrased this in the revised manuscript to be more precise. Typical planetary boundary heights at site D47 varies between 200 – 400 m.

Please note that our primary goal was to correct the variation of snowdrift flux as function of the near surface wind speed, the goal was not to get the best agreement with the snow drift observations on the expense of the relation between wind speed and snow drift flux. Since, the blowing snow flux now 'varies' in the expected, 'power-law' fashion we believe it provides a better overall representation of the fluxes.
It is worth mentioning here that, we are not aware of any other dataset of snow drift observations in Antarctica of such high quality, therefore this is the best comparison that we could do. Furthermore,

as snow drift fluxes are increasing exponentially with the wind speed, big errors are potentially made in high-windspeed regions. The impact of snowdrift sublimation in regions with low wind speeds is inherently small, even though these regions are much larger in size.

A colourblind-friendly version of the plot is included in the revised manuscript.

By differences, we meant the different simulation time periods 2010-2020 (CRYOWRF), while 2000-2012 for RACMO. Longer simulations with RACMO2.3p2 (van Wessem et al) and RACMO2.4p1 (van Dalum et al 2024) show that the Antarctic climate of 2013-2020 is very comparable to that of the period 2000-2012. Therefore, we mentioned that they are still comparable. We have rewritten this in the revised manuscript.

Colourblind-friendly versions of the plots are included in the revised manuscript.

Palm et al. (2018) results are mostly available for cloud-free skies and blowing snow layer height greater than 30 m. Since our main goal was to compare the results with the observations from site D47, we initially did not think about filtering blowing snow fluxes to only 'cloud-free' days and blowing snow layers of height greater than 30 m. Since we did not store the multi-level data required for filtering results, it would be difficult to ascertain the reason for the slight shift. Therefore, we decided to only look at the seasonal patterns and 'qualitatively' compare our results since it was not our goal to perform a one-to-one comparison of our results with the satellite observations. We will mention the limitation in the revised manuscript in detail.

Fixed in the revised manuscript.

Antarctic winter months (April–September), summer months (October–March). Rewritten in the revised manuscript.

It is mentioned in detail in Table 4.6, there is a 41% increase in the blowing snow sublimation with RpNew when compared to Rp3. Rewritten in the revised manuscript.

The main reason is to present the results of the average over a decade, so we decided to leave 2011-2012 from the Yearly averages. In the revised manuscript we have included the averages between 2000–2012, to avoid confusion.

The main reason is to present the results of the average over a decade, so we decided to leave 2011-2012 from the Yearly averages. In the revised manuscript we have included the averages between 2000–2012, to avoid confusion.

- L. 534: I am again confused by the time frame of the experiments. Here, the period 2000-2010 is mentioned while in l. 249 values are reported for the period 2010-2012. Please clarify.

The main reason is to present the results of the average over a decade, so we decided to leave 2011-2012 from the Yearly averages. In the revised manuscript we have included the averages between 2000–2012, to avoid confusion.

**Response to RC2**

**Reviewer:** The authors introduce a new version of the blowing snow scheme in RACMO. The new model is described and a first validation comparing the model to local as well as satellite blowing snow measurements is provided. Furthermore, the authors also discuss their results in comparison to an older version of RACMO and another continent-wide simulation over Antarctica. The paper overall fits the scope of the journal. Below a list of comments if provided, which should be taken into account before considering the paper for publication.

**Response:** We thank the reviewer for spending time on reviewing our manuscript. Below, we indicate the changes made in the revised manuscript.

General / major comments:

For the validation against measurements as well as compared to the other model, the term "significant" is used regularly in the text. It is unclear if significance tests have been performed. Are the changes actually significant based on significance tests? And if yes, which changes are and which are not? If this is not feasible at least the wording should be adapted accordingly (see detailed comments below).

For all the linear regressions presented in the manuscript, p-value was used to determine if the correlations are significant. We have modified the manuscript to include p-value of all regressions. We will also adapted the manuscript wherever necessary.

The authors state, that there are differences between the satellite based estimate of blowing snow frequency and the model based results due to the fact that on one hand the satellite based results only start at 30 m above ground, while the results for RACMO include all the layers close to the surface. Furthermore, CALIPSO can only see blowing snow during cloud-free conditions or for optically thin clouds. Did the authors consider excluding cloudy days and the model levels below 30 m for the analysis? What is the reason for not applying these corrections to the displayed result? This would give the comparison more weight, as then the results could more directly be compared (relates to P16, L380ff).

Main goal of our study was to fix the numerical artefact in the variation of blowing snow fluxes with wind speed and evaluate it by comparing the blowing snow results with the observations from D47, Antarctica. Therefore, we did not store the multi-level blowing snow transport data (or filtered data above 30 m). After the runs were done, we decided to do a 'qualitative' comparison with Palm et al. (2018). Therefore, we had to only look at the seasonal patterns and 'qualitatively' compare our results, since it was not our goal to perform a one-to-one comparison of our results with the satellite observations. Since Palm et al. (2018) [6] observations are limited to mostly optically thin cloud conditions, we believe the observations do not give the complete picture of the blowing snow events, and we intended to report the blowing snow events as we see from the model results. Therefore, with the current data output, it is impossible to perform the said filtering. Nonetheless, we tried to filter the blowing snow flux with the cloud fraction, but as expected, this yielded high blowing snow frequency as it is the total integrated blowing snow flux from all the layers.

We have clarified this in the revised manuscript.

Minor comments:
P1, L7: In "a non-uniformly discretized ice particle radii to . . . " radii - radius
Fixed in the revised manuscript.

P4, L85: TrasnfEr - Transfer
Fixed in the revised manuscript.

P5, L113: empirircal - empirical
Fixed in the revised manuscript.

P5, L114: ", is" does not seem to fit the sentence. Remove ", is" or reformulate.
Reformulated in the revised manuscript.

P9, L223: Is wind speed measured at 2m above ground? Based on Amory, 2020 it seems to be at 2.8 m
originally for station D-47. Is 2m an average over the measurement period given the elevation changes
due to snow? (if a change is required take it into account accordingly for P10, L261).
Your observation is correct that the 'original' height of the sensor at D47 is 2.8 m. However, in the paper
they mention that, due to harsh weather conditions at D47, they could not do reset the height of the
sensors. As a result, by late December 2012, the measurement heights decreased from their initial values
to 1.5 m for wind speed and direction and 0.9 m for temperature and relative humidity. The period we
considered for evaluation i.e. 2011, the height was reduced from its initial 2.8 m and was lower. And the
processed data that is made available on Zenodo [5], the height of the measurement is mentioned as 2
m. Furthermore, in the supplementary materials of the data paper [4] they mention that mean values for
wind speed, temperature, and relative humidity are determined from the measurement level closest to 2
m.
We have clarified in the revised manuscript.

P11: In the comparison between the modelled and the measured data, it is shown that the results improve
when simulating with blowing snow and even more when using the new version. Especially in lines 270ff
the authors talk about significant biases and improvements. Are these statements based on a significance
test? What kind of test was used? – If not either the wording should be changed or a significance test
should be performed to confirm. In general, I think it would be interesting to see which improvements
are significant.
For all the linear regressions presented in the manuscript, all the regression coefficients are tested for sta-
tistical significance using the p-value. Specifically, for all the regressions p-value $< 0.01$. We will modify
the manuscript to specify the p-value of regressions. We have rephrased this in the revised manuscript to
avoid confusion.

P14, Section 4.1.3: It is discussed that Rp3 overall shows a better performance in predicting the blowing
snow frequency overall, while RpNew mainly improves the prediction of higher magnitude blowing snow
events. It is great to see that RpNew improves for the higher magnitude events. It would, however, be
interesting to see both effects discussed? Why is there an improvement for the higher magnitude events
and why does it perform worse overall? What are the impacts of the decrease in performance when
looking at the overall blowing snow frequency.
We have added a small discussion on why RpNew predicts comparatively lower number of events.
This is due to the comparatively 'better' performance of Rp3 during the Antarctic summer. Specifically,
in Rp3, instead of using the friction velocity $u_*$, calculated in the land surface scheme, $u_*$ was recalculated
in every step with a simple log-law assumption (with neutral stratification). Since the assumption was

[Figure]

Figure 1: Threshold friction velocity for Rp3 and RpNew. Time is shown in month.

not correct, we changed it and used the friction velocity calculated from the physics module. Simply put, the assumption which was previous used (though not correct) reduced the threshold friction velocity, and thereby we observed more blowing snow events with Rp3. Figure 1 shows the threshold friction velocity used in Rp3 and RpNew, it is clear from the figure that assumptions and simplifications made in Rp3 reduced the threshold friction velocity overall which would result in higher blowing snow events compared to RpNew.
We have clarified this in the revised manuscript.

P15, L367: "... 5(a)).This analysis..." - Missing space after.
Fixed in the revised manuscript.

P21, L476: "...in under-prediction blowing snow particle concentration." - Should probably read: "...in an under-prediction of the blowing snow particle concentration."
Fixed in the revised manuscript.

P22, Table 4: From just reading the table caption and the formula, it is not clear that ERds only refers to the eroded snow blown off the continent. Please clarify.
ERds only considers the transport aspect of snowdrift. ERds is positive in case of erosion due to divergence of the snowdrift flux, and negative if convergence of the snowdrift flux brings snow to a grid box. So when ERds is negative at but less than the snow drift sublimation, snow drift still has a net erosion effect.
Furthermore, as ERds only considers the snow redistribution, the spatially integrated impact on the SMB is zero as long as drifting snow is not blown of the ice sheet. That is the reason why ERds is locally very important, but integrated over the ice sheet insignificant.
We have clarified this in the revised manuscript.

P22, Table 4: For "(a) Difference between RpNew and RP3, and (b) SMB difference between RpNew (2000-2010) with CRYOWRF (2010–2020)": Add information about the two differences shown. E.g. ... in Gt (% of xxx). To which value is the percentage related?
It is the percentage change in the quantities compared to previous version i.e. Rp3. It is calculated as the (RpNew- Rp3)/(Rp3). The word 'difference' might not be accurate here, we have rewritten this as % change in the revised manuscript.

P22, Table4b and P23, L509ff: The time periods of the RACMO and CRYOWRF simulations are not the same (RACMO: 2000-2010; CRYOWRF: 2010-2020). The difference in the between the simulated

[Figure]

Figure 2: Variation of SMB and its components from RACMO2.4, newest version of RACMO with additional physics update. The figure is a part of the manuscript in preparation.

Our experience with RACMO runs suggest that the total sublimation does not vary much in the decade between 2000-2020. Therefore, we do not expect a large difference in the results, especially sublimation, during this time period.

To elaborate, after the submission of the current manuscript, the blowing snow updates along with other Physics updates were introduced in RACMO, version 2.4p1 [7]. Figure 2 shows the yearly average of SMB components for RACMO 2.4p1. The figure is a part of the manuscript in preparation. As can be seen from the figure the sublimation (SUSD) predicted by RACMO2.4p1 does not vary much between 2000-2020, and therefore we think it is justified to compare the results.

We have mentioned the differences in the time periods and their possible influence in the revised manuscript.

This can be further explored by comparing the blowing snow sublimation for RpNew or Rp3 with NO-DRIFT case. We focused more on comparing RpNew and Rp3 in the initial version of the manuscript and therefore, we did not include the surface sublimation plots for the NO-DRIFT case in Figure 9. In the revised manuscript, we will add additional discussion related to this.

On observing this trend with CRYOWRF, it would be difficult to show this without a simulation of CRYOWRF with the blowing snow model switched-off. However, the data provided by Gerber et al. (2023) does not include a simulation without the blowing snow sublimation. However, we will include monthly sublimation from CRYOWRF in our plots (see Figure 3), from the figure we see that major difference between Rp3 and RpNew comes in winter months when blowing snow sublimation is dominant. Increase in blowing snow sublimation of around 6 Gt/yr in winter is balanced by the 'condensation' of approximately 6 Gt/yr. As you can see in the figure, CRYOWRF does not show such a 'condensation' as the surface sublimation during winter from CRYOWRF is nearly zero. This indicates that such an effect

[Figure]

Figure 3: Monthly variation of surface-, blowing snow, and total sublimation.

is not present within CRYOWRF. But, for a definitive answer, we would need a simulation of CRYOWRF without the blowing snow model, and can be explored in the future.
We have added a discussion related to this in the revised manuscript.

P23, L518: The authors state that the difference in erosion due to snow being blown off Antarctica is "significant". Is this based on a significance tests? What test has been used? Please specify or reformulate.
We have reformulated this sentence in the revised manuscript.

P23, L527: I think it should read "radius classes" instead of "radii classes"
Fixed in the revised manuscript.

P24, L536: "insitu observation from site D47": observation -¿ observations
Fixed in the revised manuscript.

P25, L570: The authors state that the RACMO data is available, but they don't say where. Please add a link to the repository from which the data can be downloaded.
A link to the dataset has been added in the revised manuscript.

**References**

[1] J. T. M. Lenaerts, M. R. van den Broeke, S. J. Déry, E. van Meijgaard, W. J. van de Berg, S. P. Palm, and J. Sanz Rodrigo. Modeling drifting snow in antarctica with a regional climate model: 1. methods and model evaluation. *Journal of Geophysical Research: Atmospheres*, 117(D5), 2012.

[2] H. Gallée, G. Guyomarćh, and E. Brun. Impact of snow drift on the antarctic ice sheet surface mass balance: possible sensitivity to snow-surface properties. *Boundary-Layer Meteorology*, 99:1–19, 2001.

[3] S. J. Déry and M. K. Yau. A bulk blowing snow model. *Boundary-Layer Meteorology*, 93:237–251, 1999.

[4] C. Amory. Drifting-snow statistics from multiple-year autonomous measurements in adélie land, east antarctica. *The Cryosphere*, 14(5):1713–1725, 2020.

[5] C. Amory, C. Genthon, and V. Favier. A drifting snow data set (2010-2018) from coastal adelie land, eastern antarctica. *Data*, 2020.

[6] S. P. Palm, V. Kayetha, and Y. Yang. Toward a satellite-derived climatology of blowing snow over antarctica. *Journal of Geophysical Research: Atmospheres*, 123(18):10–301, 2018.

[7] Christiaan T van Dalum, Willem Jan van de Berg, Srinidhi N Gadde, Maurice van Tiggelen, Tijmen

van der Drift, Erik van Meijgaard, Lambertus H van Ulft, and Michiel R van den Broeke. First results of the polar regional climate model racmo2. 4. *EGUsphere*, 2024:1–36, 2024.

---

## Referee Report (RR1)

The authors Gadde and van de Berg improved their manuscript entitled "Contribution of blowing snow sublimation to the surface mass balance of Antarctica" after implementing the comments from both reviewers. The authors shortened many parts in the manuscript which makes the text more precise and easier to read and follow. Overall, I would like to see a short discussion and the author's thoughts on future model developments and how to further improve the representation of blowing snow sublimation and surface sublimation in future model versions.

After considering the minor comments below, I am suggesting this manuscript for publication in *The Cryosphere*.

- The abstract is quite detailed and long which is fine if all information is relevant. I am wondering, however, if this part could be shortened and more precise on the main findings of the paper. In line 3, you write "among other things". What do you mean by that?
- L. 62: speed winds --> wind speeds
- L. 253: Please guide the reader and describe the mismatch between blowing snow form Rp3 compared to the observations. It might be obvious for you, but I would prefer to read more description than "It is evident from the figure...".
- L. 263: Are you going to analyse the seasonal difference with the CALIPSO data later in the manuscript? Then please refer to the respective part of the manuscript.
- Table 1: From the caption, it is not clear to me what the two numbers in each cell mean. Please provide a precise table caption.
- L. 352 and Figure 6: Please indicate what data you are showing here (RpNew or Rp3).
- L. 362f.: Please be more specific when saying 'qualitatively similar'. Can you provide any numbers or figures for that? Currently, the reader has to entirely trust you that there is a similarity between your model runs and the CRYOWRF simulations.
- L. 412: can you quantify the contribution of blowing snow sublimation to the total sublimation in winter?
- L. 427: What could be a reason why CRYOWRF does not show a negative surface sublimation? Can you elaborate on this?
- Table 2b: how do you compare RpNew values from 2000-2012 with CRYOWRF from 2010-2020? How representative is this comparison? You mention in l. 449 and 450 that RACMO does not vary much in the decades between 2000 and 2020. I am wondering what is then driving the inter-annual variability in the model if there is no difference between the decade from 2000 to 2010 and the decade from 2010 to 2020.
- L. 460: are you sure that sublimation is the largest ablation term compared to runoff and ice melt considering the entire Antarctic ice sheet?

---

## Author Response (AR2)

**Response to Reviewer**

**Reviewer:** The authors Gadde and van de Berg improved their manuscript entitled "Contribution of blowing snow sublimation to the surface mass balance of Antarctica" after implementing the comments from both reviewers. The authors shortened many parts in the manuscript which makes the text more precise and easier to read and follow. Overall, I would like to see a short discussion and the author's thoughts on future model developments and how to further improve the representation of blowing snow sublimation and surface sublimation in future model versions.

We thank the reviewer for spending time on reviewing our manuscript. Below we explain the changes to the manuscript. While this manuscript was under review, RACMO2.4 was under development and we refer to [1] for the most recent model updates. In RACMO2.4 no further improvement on blowing snow sublimation or surface sublimation is implemented. In the current version of the blowing snow model, the snow properties such as dendricity and helicity are assumed to be constant ($d = 0.5$ and $s = 0.5$). The variance in the observational data is not captured accurately by the model due to the aforementioned simplification, it will be a major model improvement to fix this simplification. In addition, blowing snow model is not coupled with the radiation model, and as such the effect of blowing snow on radiative transfer is not considered, which will be looked into in the future. Finally, we intend to improve the surface snow density and wind packing, as we have observed that the poor representation of the temporal and spatial variability of the surface snow density is giving the largest uncertainties and model deviations in the modelled firn densification and firn air content.

**Reviewer:** After considering the minor comments below, I am suggesting this manuscript for publication in The Cryosphere.

The abstract is quite detailed and long which is fine if all information is relevant. I am wondering, however, if this part could be shortened and more precise on the main findings of the paper. In line 3, you write "among other things". What do you mean by that?

**Response:** We agree and we have shortened the length of the abstract. By 'among other things' we meant the other major changes we introduced to streamline the coupling of blowing snow, which is explained in section 2.3. We have removed the phrase for clarity.

L. 62: speed winds –¿ wind speeds
Fixed in the revised manuscript.

L. 253: Please guide the reader and describe the mismatch between blowing snow form Rp3 compared to the observations. It might be obvious for you, but I would prefer to read more description than "It is evident from the figure...".
Fixed in the revised manuscript. We have explained how the peaks are not captured properly with Rp3 and also the trends of windspeed vs blowing snow fluxes are not following the expected trend.

L. 263: Are you going to analyse the seasonal difference with the CALIPSO data later in the manuscript? Then please refer to the respective part of the manuscript.
We only qualitatively compare the seasonal variation of blowing snow frequency with the CALIPSO data and as such we did not mention this in L.263 where we discuss blowing snow flux magnitude.

Table 1: From the caption, it is not clear to me what the two numbers in each cell mean. Please provide a precise table caption.
We have added the caption in revised manuscript. The first table presents the blowing snow events as simulated by RpNew and Rp3, respectively. The second table presents high blowing snow flux events calculated as $Q_T > 0.05$ kg m$^{-2}$ s$^{-1}$.

L. 352 and Figure 6: Please indicate what data you are showing here (RpNew or Rp3).
We are referring to RpNew, we will mention this in the revised manuscript.

L. 362f.: Please be more specific when saying 'qualitatively similar'. Can you provide any numbers or figures for that? Currently, the reader has to entirely trust you that there is a similarity between your model runs and the CRYOWRF simulations.
We have added a description to this. Gerber et al. (2023) [2] report a zone of strongest blowing snow along the coast of East Antarctica with highest values of blowing snow frequency slightly inland. Zones with reduced blowing snow are found over the Amery ice shelf, toward the western Queen Maud land. RpNew shows similar qualitative trends.

L. 412: can you quantify the contribution of blowing snow sublimation to the total sublimation in winter?
For RpNew blowing snow sublimation contributes 108 Gt during winter out of the total sublimation of 175 Gt $yr^{-1}$. This amounts to 62% of the total blowing snow sublimation.

- L. 427: What could be a reason why CRYOWRF does not show a negative surface sublimation? Can you elaborate on this?
Unfortunately, we do not know the reason why CRYOWRF does not show the negative surface sublimation, and such an analysis is out of the scope of current study. As CRYOWRF is one of the few models with blowing snow parameterization applicable to entire continent (aside from MAR), future intercomparisons with long-term simulaitons are necessary to obtain conclusive answers to model differences.

Table 2b: how do you compare RpNew values from 2000-2012 with CRYOWRF from 2010-2020? How representative is this comparison? You mention in l. 449 and 450 that RACMO does not vary much in the decades between 2000 and 2020. I am wondering what is then driving the inter-annual variability in the model if there is no difference between the decade from 2000 to 2010 and the decade from 2010 to 2020.
In our previous reply to the reviewer's comments we mentioned that the inter-annual variability in the blowing snow sublimation from RACMO does not vary much in the decades between 2000–2020. Therefore, we could compare the blowing snow sublimation; we did not mention that SMB or precipitation does not vary between 2000–2010 and 2010–2020. There is a large inter-annual variability in precipitation (see Figure 2 from our previous reply).

L. 460: are you sure that sublimation is the largest ablation term compared to runoff and ice melt considering the entire Antarctic ice sheet?
Yes, we are sure. See, for example, van Wessem et al. (2018) [3], Table 2. Sublimation exceeds melt, and all snow melt water refreezes in the snowpack – so it does not contribute to actual mass loss (thus ablation) term.

**References**

[1] Christiaan T van Dalum, Willem Jan van de Berg, Srinidhi N Gadde, Maurice van Tiggelen, Tijmen van der Drift, Erik van Meijgaard, Lambertus H van Ulft, and Michiel R van den Broeke. First results of the polar regional climate model racmo2. 4. *EGUsphere*, 2024:1–36, 2024.

[2] Franziska Gerber, Varun Sharma, and Michael Lehning. Cryowrf—model evaluation and the effect of blowing snow on the antarctic surface mass balance. *Journal of Geophysical Research: Atmospheres*, 128(12):e2022JD037744, 2023.

[3] M. J. van Wessem, W. J. van de Berg, P. Y. Brice Noël, E. van Meijgaard, C. Amory, G. Birnbaum, C. L. Jakobs, K. Krüger, J. T. M. Lenaerts, S. Lhermitte, et al. Modelling the climate and surface

mass balance of polar ice sheets using racmo2–part 2: Antarctica (1979–2016). *The Cryosphere*, 12(4):1479–1498, 2018.